# Sentinel-VLA: A Metacognitive VLA Model with Active Status Monitoring for Dynamic Reasoning and Error Recovery

**Wenhao Li** [1] **Xiu Su** [2] **Dan Niu** [3] **Yichao Cao** [2] **Hongyan Xu** [2] **Zhe Qu** [2] **Lei Fan** [4] **Shan You** [5] **Chang Xu** [1]

## Abstract

Vision-language-action (VLA) models have advanced the field of embodied manipulation by harnessing broad world knowledge and strong generalization. However, current VLA models still face several key challenges, including limited reasoning capability, lack of status monitoring, and difficulty in self-correction. In this paper, we introduce **Sentinel-VLA**, a metacognitive VLA model equipped with an active "sentinel" module to monitor real-time execution status. Only when necessary, such as during initial planning or upon detecting an error, the model triggers a dynamic reasoning or formulate error recovery solutions. This on-demand reasoning mechanism ensures robust decision-making while minimizing computational overhead. Notably, all training data (spanning 44 tasks and over 2.6 million transitions) is automatically generated and annotated through our designed pipeline. We also propose the Self-Evolving Continual Learning (SECL) algorithm, which allows Sentinel-VLA to identify its capability boundaries and automatically collect data for expansion, paired with Orthogonal Continual Adapter (OC-Adapter) to constrain parameter updates to an orthogonal space, thereby preventing catastrophic forgetting. Real-world experiments demonstrate that Sentinel-VLA boosts the task success rate by over 30% compared to the SOTA model, PI0.

## 1 Introduction

Large Language Models (LLMs) (Touvron et al., 2023; Bai et al., 2023; Team, 2024; Zhang et al., 2024; Zeng et al., 2026; Tan et al., 2025) and vision-language models (VLMs) (Zhu et al., 2023; Wang et al., 2024; Liu et al., 2024a; Team

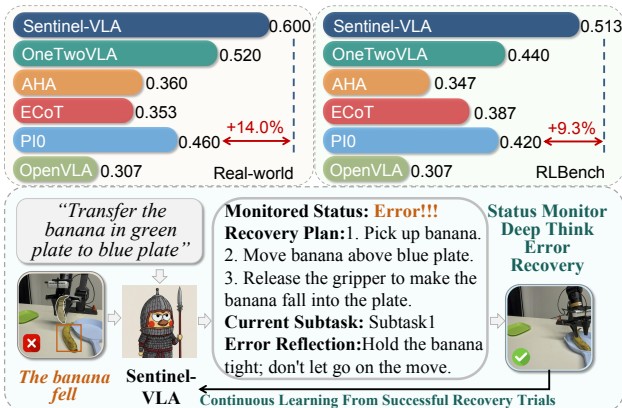

*Figure 1.* The performance and mechanism of Sentinel-VLA.

et al., 2024a; Li et al., 2025b; Wang et al., 2025; Li et al., 2025a) have recently achieved revolutionary success in the digital world. However, the path toward true AGI (Wang & Sun, 2025; Yenduri et al., 2025; Bikkasani, 2025; Restrepo, 2025) demands agents that transcend the virtual realm to interact physically with the world. Consequently, Embodied AI (Liu et al., 2025a;b; Fung et al., 2025; Feng et al., 2025; Li et al., 2025c;d; Chen et al., 2026; Li et al., 2021; 2023) has emerged as a critical frontier in the field. In this pursuit, VLA models (Kim et al., 2024; Liu et al., 2024c; Li et al., 2024b; Team et al., 2024c; Black et al., 2024; Xu et al., 2025b;a; Yang et al., 2026; Li et al., 2026a;b), which evolved from the powerful foundations of LLMs and VLMs, have rapidly become the dominant paradigm in embodied manipulation. Leveraging their extensive world knowledge, strong generalization capabilities, and robust multi-modal understanding, VLA models hold immense promise for the development of general-purpose robotic assistants.

Despite this promise, a critical gap persists between the capabilities of current VLA models and the demands of complex, long-horizon tasks in the real world. Specifically, they face three core challenges: *i) Insufficient Reasoning Capability.* Unlike LLMs and VLMs which are capable of deep reasoning, most VLA models function as direct input-to-action mappings, limiting their ability to handle complexity (Zawalski et al., 2024). *ii) Lack of Status Monitoring.* Current VLA models are unaware of runtime errors (like grasping an empty kettle when tasked with pouring water) and tend to continue execution despite being in a faulty state (Zhou

---

[1]University of Sydney [2]Central South University [3]Southeast University [4]University of New South Wales [5]Sensetime Research. Correspondence to: Xiu Su, Chang Xu <xiusu1994@csu.edu.cn, c.xu@sydney.edu.au>.

*Proceedings of the 43rd International Conference on Machine Learning*, Seoul, South Korea. PMLR 306, 2026. Copyright 2026 by the author(s).

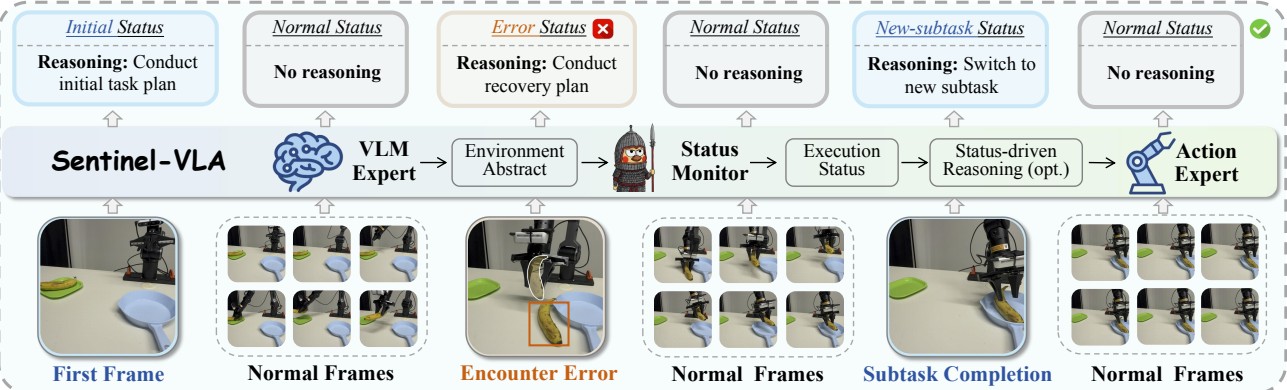

*Figure 2.* The core idea of Sentinel-VLA: In most frames, it determines a "normal status" and directly outputs an action without reasoning. In the few frames, Sentinel-VLA assesses current status and reasons as needed, generating a better action or recovering from an error.

et al., 2025a). *iii) Inability to Self-Reflect and Correct.* VLA models are incapable of learning or recovering from their mistakes (Li et al., 2024a), which severely compromises their reliability and safety in practical applications.

Existing solutions for these issues remain piecemeal and limited. For reasoning, methods like ECoT (Zawalski et al., 2024) and CoT-VLA (Zhao et al., 2025) have improved action generation with intermediate thoughts, but their rigid "reason-at-every-step" strategy is inflexible and incurs high latency. While OneTwoVLA (Lin et al., 2025) attempted selective reasoning, its reliance on a special token with randomness is unstable and lacks interpretability. Furthermore, it cannot think and act concurrently. For error handling, most existing approaches (Xia et al., 2025; Dai et al., 2024; Duan et al., 2024; Pan et al., 2025; Zhou et al., 2025a; Li et al., 2024a) rely on external models or tools for monitoring and recovery. Besides being architecturally cumbersome and incurring high overhead, the performance of such methods is also bottlenecked by the VLA's own inability to follow the complex external corrections (Glossop et al., 2025).

In this paper, we introduce **Sentinel-VLA**, a metacognitive VLA model designed with an integrated cognitive architecture. The cornerstone of Sentinel-VLA is an active monitoring module, which acts as a sentinel to vigilantly track the real-time execution status of a task. This mechanism enables a paradigm of on-demand reasoning: only when necessary—during initial planning or upon anomaly detection—does the model trigger deeper cognitive processes for reasoning or error recovery. This approach enables the agent with robust decision-making and self-correction capabilities while circumventing the high computational overhead of previous static, "reason-at-every-step" strategies.

To train this status-aware behavior, we developed EC-Gen, a scalable data generation pipeline that automatically synthesizes diverse error recovery trajectories with rich annotations, obviating the need for manual data collection. Furthermore, to ensure Sentinel-VLA continuously improves from its experiences, we propose the SECL algorithm, com-

plemented by OC-Adapter. This allows Sentinel-VLA to efficiently learn from novel real-world interactions, progressively expanding its knowledge boundary and the scope of errors it can handle, all while effectively mitigating catastrophic forgetting. Extensive experiments demonstrate that Sentinel-VLA achieves relative improvements of over 22% and 30% on RLBench (unseen tasks) and real-world settings, respectively, compared to the SOTA VLA PI0 (Black et al., 2024). To sum up, our contributions are as follows:

- We propose *Sentinel-VLA*, a unified VLA model that integrates an active status monitor for dynamic, on-demand reasoning and error recovery. This metacognitive approach enables robust reasoning and self-correction while minimizing computational overhead.

- We develop EC-Gen, a scalable pipeline that automatically synthesizes a large (2.6M+ transitions) error recovery trajectories with rich annotations. It effectively trains the model's status-aware behaviors without relying on laborious manual data collection and annotation.

- We introduce the Self-Evolving Continual Learning algorithm, featuring an Orthogonal Continual Adapter. This framework allows Sentinel-VLA to continuously learn from experiences, expanding its knowledge and capabilities while mitigating catastrophic forgetting.

**Conflict of Interest Disclosure** None.

## 2 Related Work

**Deep Think in VLA Models.** To enhance reasoning in VLA models, approaches like ECoT (Zawalski et al., 2024), CoT-VLA (Zhao et al., 2025), RoboMamba (Liu et al., 2024b), and PI0.5 (Black et al., 2025) predict intermediate auxiliary information (e.g., sub-tasks or gripper poses). However, these static outputs fail to adapt to dynamic task contexts. Recent models, including ChatVLA (Zhou et al., 2025c), ChatVLA2 (Zhou et al., 2025b), and Hume (Song et al., 2025), introduce free-form natural language thought but lack status monitoring. Consequently, they rigidly engage in latency-heavy reasoning at every step, making them

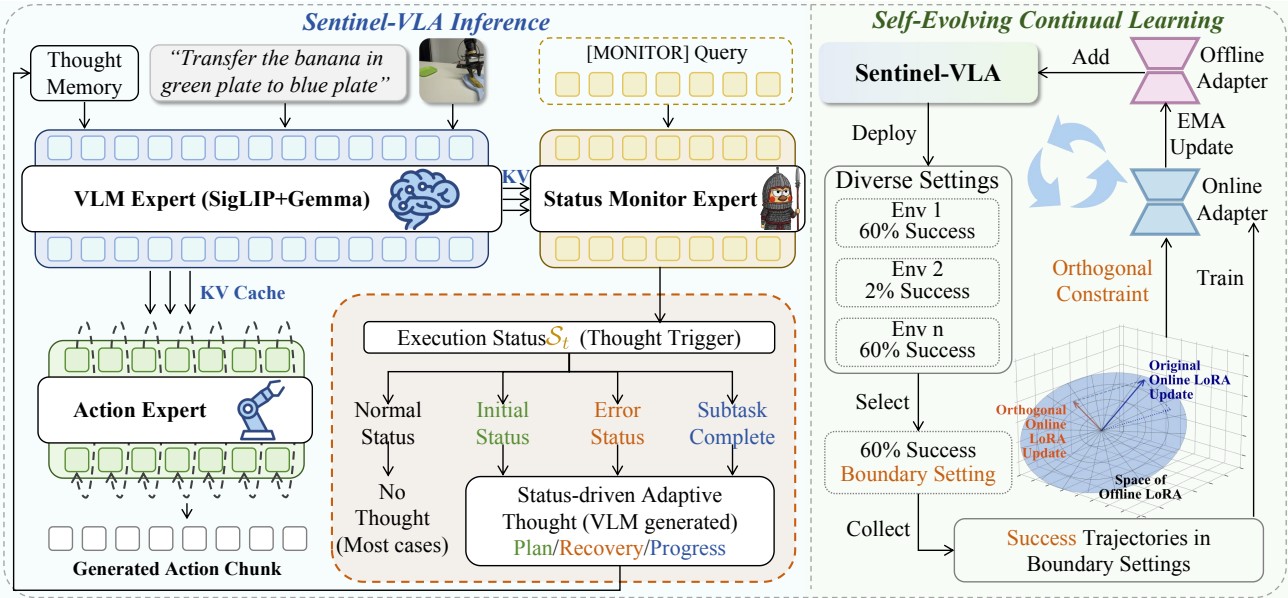

*Figure 3.* **Left:** Pipeline of Sentinel-VLA. The Status Monitor Expert activates on-demand Adaptive Thought. **Right:** Pipeline of SECL. The model continually evolves by learning from boundary success trajectories, updating its adapter with Orthogonal Constraint.

unsuitable for real-time manipulation. While OneTwoVLA (Lin et al., 2025) attempts to regulate reasoning frequency, its reliance on a stochastic token prediction renders it unreliable and difficult to interpret.

**Error Recovery in VLA Models.** To address the critical lack of error resilience in VLAs, researchers have introduced external oversight modules. Works such as AHA (Duan et al., 2024), Phoenix (Xia et al., 2025), and Racer (Dai et al., 2024) utilize external LLMs/VLMs for corrective feedback, while others employ rule-based monitors (Zhou et al., 2025a), specialized tools (Li et al., 2024a), or 3D discrepancy detection (Pan et al., 2025). Despite their utility, these methods suffer from architectural rigidity and a fundamental bottleneck: VLAs often struggle to precisely follow external instructions (Glossop et al., 2025), meaning even valid recovery plans from external monitors may not be correctly executed.

## 3 Sentinel-VLA

The architecture of Sentinel-VLA comprises a VLM expert ($E_{vlm}$), an action expert ($E_{act}$), and a novel Status Monitor expert ($E_{sm}$). These components share attention mechanisms for efficient information exchange, as depicted in Figure 3.

### 3.1 Dynamic Reasoning via Active Status Monitor

**Active Monitoring.** At each timestep $t$, the model receives the current image observation $\mathcal{I}_t$ and the task instruction $\mathcal{T}$. The VLM expert projects this input into a latent feature space, yielding the key-value cache context $\mathcal{K}_t, \mathcal{V}_t = E_{vlm}(\mathcal{I}_t, \mathcal{T})$. Then, to determine the demand of reasoning, the Status Monitor expert $E_{sm}$ acts as a "sen-

tinel" to assess current status. $E_{sm}$ first processes a learnable [MONITOR] query through its first layer's Q matrix, producing the query vector $Q_{sm}$. This vector then probes the internal $(\mathcal{K}_t, \mathcal{V}_t)$ cache within the VLM expert via cross-attention, allowing $E_{sm}$ to determine if the current context requires the activation of a specific cognitive capability. The output of $E_{sm}$ is passed through an MLP head to compute the trigger probability distribution:

$$\mathcal{S}_t = \text{softmax}(\text{MLP}(E_{sm}(Q_{sm}, \mathcal{K}_t, \mathcal{V}_t = E_{vlm}(\mathcal{I}_t, \mathcal{T}))))$$
(1)

To introduce task planning and error recovery capabilities, we define a set of trigger states, $\mathcal{S}_t$, corresponding to the model's core cognitive functions.

$$\mathcal{S}_t \in \{\text{Initial}, \text{Normal}, \text{New-subtask}, \text{Error}\} \quad (2)$$

**Dynamic Reasoning.** Conditioned on the activated trigger status $\mathcal{S}_t$, the VLM expert $E_{vlm}$ is invoked again to determine whether and how to perform deep thinking. We maintain a dynamically updated thought memory, $\mathcal{M}_t$, which stores information such as the task plan, the current subtask, and error reflections. Let $\Psi_{reason}(\cdot)$ denote the cognitive policy executed by $E_{vlm}$. The update logic for $\mathcal{M}_t$ is:

$$\mathcal{M}_t = \begin{cases} \Psi_{plan}(\mathcal{I}_t, \mathcal{T}) & \text{if } \mathcal{S}_t = \text{Initial} \\ \mathcal{M}_{t-1} & \text{if } \mathcal{S}_t = \text{Normal} \\ \Psi_{update}(\mathcal{M}_{t-1}, \mathcal{I}_t) & \text{if } \mathcal{S}_t = \text{New-subtask} \\ \Psi_{recover}(\mathcal{M}_{t-1}, \mathcal{I}_t) & \text{if } \mathcal{S}_t = \text{Error} \end{cases}$$
(3)

Here, $\Psi_{plan}$ produces an initial task plan $\mathcal{P} = \{p_i\}_{i=1}^N$, where each $p_i$ is a subtask, and sets the current subtask to

$p_1$. $\Psi_{update}$ advances the current subtask in memory from $p_i$ to $p_{i+1}$. $\Psi_{recover}$ formulates a new recovery plan $\mathcal{P}_{rec}$, a new current subtask, and an error reflection. Critically, in the "Normal" state, which constitutes the vast majority of execution time, no specific capability is triggered, and the model generates no new thoughts ($\mathcal{M}_t = \mathcal{M}_{t-1}$). It directly leverages the existing thought memory for decision-making, significantly reducing computational overhead.

**Act with Reasoning.** Finally, the action expert $E_{act}$ integrates all context to generate the final action $A_t$:

$$A_t = E_{act}(\mathcal{I}_t, \mathcal{T}, \mathcal{S}_t, \mathcal{M}_t) \tag{4}$$

**Training with Unified Objective.** The overall training loss for this dynamic chain-of-thought process, $\mathcal{L}_{DCoT}$, combines a flow matching loss for action prediction, $\mathcal{L}_{flow}$, a cross-entropy loss for thought generation, $\mathcal{L}_{thought}$, and a cross-entropy loss for the status monitor, $\mathcal{L}_{monitor}$:

$$\mathcal{L}_{DCoT} = \mathcal{L}_{flow} + \lambda(\mathcal{L}_{thought} + \mathcal{L}_{monitor}) \tag{5}$$

where $\lambda$ is a hyperparameter. Sentinel-VLA is optimized in a fully end-to-end manner, ensuring strict alignment between metacognitive judgment and physical execution.

### 3.2 EC-Gen Data Generation Pipeline

To train the status-aware behaviors of Sentinel-VLA, we develop EC-Gen, a pipeline that transforms expert trajectories into error-correction sequences via a stochastic perturbation mapping. Let a successful trajectory be a sequence of waypoints $\tau = \{w_1, \ldots, w_N\}$, where each waypoint $w_i = \langle \mathbf{e}_i, g_i \rangle$ consists of an $SE(3)$ end-effector pose $\mathbf{e}_i$ and a binary gripper state $g_i \in \{0, 1\}$. We define an error injection operator $\Phi(w_j, \epsilon)$ that maps a target waypoint $w_j$ to an erroneous state $w'_j$ based on a specific failure $\epsilon$.

$$w'_j = \langle \mathbf{e}'_j, g'_j \rangle = \Phi(\langle \mathbf{e}_j, g_j \rangle, \epsilon) \tag{6}$$

While real-world errors are diverse, we posit that the vast majority of manipulation failures can be decomposed into or combined from three core failure modalities: interaction, spatial, and semantics. We design our error generation pipeline to cover these fundamental dimensions:

**1. Object Interaction Errors ($\epsilon_{gripper}$):** This modality covers failures at the point of interaction, such as failing to grasp or release an object. At a waypoint $w_j$ where a gripper state change was supposed to occur, we suppress this change. The erroneous waypoint $w'_j$ is formed by retaining the original pose $e_j$ but adopting the gripper state from the previous waypoint:

$$w'_j = \Phi(w_j, \epsilon_{gripper}) = \langle \mathbf{e}_j, g_{j-1} \rangle \tag{7}$$

**2. Spatial Localization Errors ($\epsilon_{pose}$):** This dimension relates to incorrect spatial positioning. A deviation is introduced to the end-effector's pose. This is implemented by

---

**Algorithm 1** Inference Process of Sentinel-VLA

1: **Input:** Task instruction $\mathcal{T}$; Experts $E_{vlm}, E_{sm}, E_{act}$; Thought memory $\mathcal{M}_0 \leftarrow \emptyset$; Timestep $t \leftarrow 0$
2: **while** task is not terminated **do**
      // ——— 1. Active Capability Triggering ———
3:               ▷ Encode context with VLM expert
4:    $\mathcal{K}_t, \mathcal{V}_t \leftarrow E_{vlm}(\mathcal{I}_t, \mathcal{T})$
5:                   ▷ Process the `[MONITOR]`
6:    $Q_{sm} \leftarrow E_{sm}.\text{ProjectToQuery}(\text{[MONITOR]})$
7:         ▷ Perceive trigger status by probing VLM
8:    $\mathcal{S}_t \leftarrow \text{softmax}(\text{MLP}(E_{sm}(Q_{sm}, \mathcal{K}_t, \mathcal{V}_t)))$
      // ——— 2. Decide whether and what to think ———
9:    **if** $\mathcal{S}_t = $ Initial **then**
10:         ▷ System starts, trigger initial planning.
11:      $\mathcal{M}_t \leftarrow \text{GeneratePlan}(\mathcal{I}_t, \mathcal{T})$
12:    **else if** $\mathcal{S}_t = $ New-subtask **then**
13:        ▷ Subtask complete, trigger subtask update.
14:      $\mathcal{M}_t \leftarrow \text{UpdateSubtask}(\mathcal{M}_{t-1}, \mathcal{I}_t, \mathcal{T})$
15:    **else if** $\mathcal{S}_t = $ Error **then**
16:        ▷ Error detected, trigger recovery reasoning.
17:      $\mathcal{M}_t \leftarrow \text{Recovery}(\mathcal{M}_{t-1}, \mathcal{I}_t, \mathcal{T})$
18:    **else**         ▷ $\mathcal{S}_t = $ Normal, skip thinking.
19:      $\mathcal{M}_t \leftarrow \mathcal{M}_{t-1}$
20:    **end if**
      // ——— 3. Action Generation ———
21:    Generate the final action using all available context:
22:    $A_t \leftarrow E_{act}(\mathcal{I}_t, \mathcal{T}, \mathcal{S}_t, \mathcal{M}_t)$
23:    $t \leftarrow t + 1$
24: **end while**

---

adding a perturbation vector $\Delta_{pose}$ to the original pose $e_j$, while keeping the gripper state $g_j$ unchanged:

$$w'_j = \Phi(w_j, \epsilon_{pose}) = \langle \mathbf{e}_j \oplus \Delta_{\text{pose}}, g_j \rangle \tag{8}$$

where $\oplus$ denotes the composition of the pose with the translational noise.

**3. Semantic Understanding Errors ($\epsilon_{sem}$):** This modality represents a failure to correctly interpret task semantics, such as acting on the wrong object. The robot interacts with the wrong object. We simulate this by shifting the end-effector's original position by the displacement vector from the intended object to an incorrect one. Let $\mathbf{p}_{target}$ and $\mathbf{p}_{distractor}$ denote the positions of the correct and incorrect objects, respectively. We define a semantic shift vector $\mathbf{v}_{shift} = \mathbf{p}_{distractor} - \mathbf{p}_{target}$. The perturbed state directs the effector towards the wrong semantic entity:

$$w'_j = \Phi(w_j, \epsilon_{sem}) = \langle \mathbf{e}_j \oplus \mathbf{v}_{shift}, g_j \rangle \tag{9}$$

For interaction and localization errors, we construct a recovery sequence by inserting an intermediate transition waypoint $w_{trans}$ followed by the original correct waypoint $w_j$,

*Figure 4.* EC-Gen pipeline of scalable data generation for error recovery trajectories.

forming a segment $\langle..., w_{j-1}, w'_j, w_{trans}, w_j, ...\rangle$. For semantic understanding errors, the recovery is direct and contains no intermediate transition waypoint, forming a segment $\langle..., w_{j-1}, w'_j, w_j, ...\rangle$.

We then automatically annotate this synthetic data with CoT labels. The task plan $\mathcal{P}$ is manually annotated once per task. Since key-waypoints naturally split subtasks, the subtask $p_i$ and status $\mathcal{S}_t$ for each frame are known. The error reflection and experience, $R_e$, are generated from a predefined template-base generator $G_{template}$ based on the injected error type $\epsilon$. Crucially, during training, the action loss $\mathcal{L}_{flow}$ is masked for the erroneous actions produced from $w'_j$ to prevent the model from learning incorrect behaviors.

### 3.3 Self-Evolving Continual Learning

The SECL algorithm aims to expanding the model's "knowledge boundary"—the frontier where success rates fluctuate between stability and failure. We conceptualize this process as an iterative expansion of the policy $\pi_\theta$ parameterized by base weights $W_{base}$ and an evolving adapter module.

**1. Deploy and Identify the Boundary:** The current stable model, with weights $W_{deploy} = W_{base} + \Delta W_{offline}$, is deployed across a range of environmental settings $\mathcal{H}$. Let $\text{SR}(W, h)$ denote the task success rate of the model with weights $W$ in setting $h$. We identify the model's current performance edge—the "boundary settings" $\mathcal{H}_{boundary}$—where this success rate is within a specific range $[\tau_{low}, \tau_{high}]$:

$$\mathcal{H}_{boundary} = \{h \in \mathcal{H} \mid \tau_{low} \leq \text{SR}(W_{deploy}, h) \leq \tau_{high}\} \tag{10}$$

Settings with a success rate below $\tau_{low}$ are considered too difficult for effective learning at this stage, while those above $\tau_{high}$ are considered already mastered.

**2. Learn from Boundary Successes:** By running the model within the identified boundary settings $\mathcal{H}_{boundary}$, we collect the set of successful execution trajectories, $D_{boundary}$. Let $\pi(h) = \text{Rollout}(W_{deploy}, h)$ represent the policy rollout distribution in setting $h$, and $S(\xi)$ be an indicator function for a successful trajectory $\xi$. The dataset is formally defined as:

$$D_{boundary} = \bigcup_{h \in \mathcal{H}_{boundary}} \{\xi \sim \pi(h) \mid S(\xi) = 1\} \tag{11}$$

This data, representing skills at the edge of the model's current competency, is used to train a new online LoRA adapter, $\Delta W_{online}$. To integrate this new knowledge without disrupting existing skills (i.e., catastrophic forgetting), we employ the OC-Adapter mechanism.

**3. Consolidate Knowledge via EMA Fusion:** After the online adapter is trained, its learned knowledge is gently merged into the long-term offline weights. This step consolidates the newly acquired skill into the model's stable memory base.

$$\Delta W_{offline} = \alpha \cdot \Delta W_{offline} + (1 - \alpha) \cdot \Delta W_{online} \tag{12}$$

The updated offline weight $\Delta W_{offline}$ is then added to $W_{base}$ for further expansion.

**Orthogonal Continual Adapter (OC-Adapter):** OC-Adapter ensures that new skills are learned in a way that is mathematically decorrelated from previously learned skills, thus minimizing interference.

In LoRA-based methods, parameter updates $\Delta W$ are confined to the column space of a low-rank matrix $B$ (i.e., $\Delta W = B \times A$). When introducing a new "online" adapter ($\Delta W_{online} = B_{online} \times A_{online}$), its updates can interfere with or overwrite knowledge learned by previous "offline" adapters ($B_{offline}$), as their column spaces may overlap.

To address this, our core innovation is to introduce an orthogonality constraint during the training of the online adapter. This term explicitly penalizes the overlap between the column space of $B_{online}$ and that of $B_{offline}$:

$$\mathcal{L}_{ortho} = \|B_{offline} \times B_{online}\|^2 \tag{13}$$

where $\| \cdot \|^2$ is the L2 norm. The total loss for training the online adapter in round $k$ is therefore a combination of the primary task loss and this orthogonality penalty:

$$\mathcal{L}_{SECL} = \mathcal{L}_{DCoT}(D_{boundary}) + \beta \mathcal{L}_{ortho} \tag{14}$$

where $\beta$ is a hyperparameter that balances task performance with knowledge preservation. This entire SECL process

*Table 1.* Comprehensive Success Rate Comparison across RLBench, LIBERO-LONG, and Real-world Tasks.

| Category | Task | OpenVLA | PI0 | ECoT | AHA+OpenVLA | OneTwoVLA | Sentinel-VLA (Ours) |
|---|---|---|---|---|---|---|---|
| **RLBench Seen** | Close box | 62% | 84% | 66% | 64% | 86% | **94%** |
| | Close laptop lid | 38% | 64% | 50% | 58% | 68% | **78%** |
| | Toilet seat down | 72% | 100% | 86% | 80% | 100% | **100%** |
| | Close fridge | 76% | 90% | 84% | 72% | 86% | **94%** |
| | Frame off hanger | 16% | 72% | 20% | 30% | 70% | **74%** |
| | Water plants | 14% | 44% | 24% | 16% | 26% | **46%** |
| | Phone on base | 20% | 26% | 24% | 18% | 32% | **40%** |
| | Put rubbish in bin | 8% | 20% | 12% | 8% | 22% | **24%** |
| | Change clock | 14% | 20% | 16% | 18% | 22% | **22%** |
| | *Average (Seen)* | 35.6% | 57.8% | 42.4% | 40.4% | 56.9% | **63.5%** |
| **RLBench Disturbed** | Close box | 46% | 70% | 56% | 60% | 74% | **86%** |
| | Close laptop | 30% | 54% | 38% | 42% | 62% | **70%** |
| | Toilet seat | 54% | 82% | 74% | 68% | 84% | **90%** |
| | Close fridge | 62% | 78% | 70% | 66% | 82% | **86%** |
| | Frame off | 10% | 52% | 10% | 24% | 54% | **64%** |
| | Water plants | 8% | 30% | 14% | 12% | 24% | **32%** |
| | Phone on base | 10% | 18% | 14% | 14% | 24% | **30%** |
| | Put rubbish | 0% | 14% | 8% | 8% | 16% | **16%** |
| | Change clock | 10% | 16% | 10% | 12% | 16% | **18%** |
| | *Average (Disturbed)* | 25.6% | 46.0% | 32.7% | 34.0% | 48.4% | **54.7%** |
| **RLBench Unseen** | Sweep dustpan | 52% | 62% | 60% | 54% | 66% | **72%** |
| | Umbrella out | 32% | 46% | 42% | 34% | 46% | **54%** |
| | Wine at rack | 8% | 18% | 14% | 16% | 20% | **28%** |
| | *Average (Unseen)* | 30.7% | 42.0% | 38.7% | 34.7% | 44.0% | **51.3%** |
| **LIBERO-LONG** | Average | 53.7% | 85.2% | 62.4% | 71.2% | 87.8% | **90.7%** |
| **Real-world** | Stack cube | 24% | 46% | 28% | 32% | 50% | **54%** |
| | Pour water | 36% | 50% | 42% | 36% | 56% | **66%** |
| | Sweep rubbish | 32% | 42% | 36% | 40% | 50% | **60%** |
| | *Average (Real)* | 30.7% | 46.0% | 35.3% | 36.0% | 52% | **60.0%** |

allows Sentinel-VLA to robustly and continually learn from experience, progressively expanding its ability to solve complex and unseen problems.

## 4 Experiments

### 4.1 Implementation Details

**Model Architecture.** Our implementation of Sentinel-VLA is built upon the publicly available PI0 model (Black et al., 2024). Both the VLM expert ($E_{vlm}$) and the action expert ($E_{act}$) are inherited from PI0, and are the 3B PaliGemma (Steiner et al., 2024) and the 330M Gemma (Team et al., 2024b), respectively. The new status monitor expert ($E_{sm}$) we have added has a network architecture consistent with the action expert.

**Training Data.** With EC-Gen, we created a large-scale pre-training dataset that consists of **11,000 trajectories**, totaling approximately **2.6 million transitions**, and covers **44 different RLBench tasks**.

**Training Setup.** The model was pre-trained for 90,000 steps on 8 NVIDIA A100 GPUs with a batch size of 128. We used the learning rate of $2.5 \times 10^{-5}$ and a cosine decay schedule. The loss weighting hyperparameters were set to $\lambda = 0.1$. For the thought generation loss, $\alpha = 0.9$ for online-offline LoRA merge and $\beta = 0.5$ for the OC-Adapter orthogonality loss during the SECL phase. $\tau_{low}$ and $\tau_{high}$ are set to 20% and 80%, respectively. For task-specific

adaptation, the pre-trained Sentinel-VLA is finetuned for 30,000 steps. Each task evaluation involves 50 execution trials to robustly calculate the average success rate.

**Compared Baselines.** We compared Sentinel-VLA against current SOTA VLA base models, error-handling and CoT reasoning methods, including PI0 (Black et al., 2024), OpenVLA (Kim et al., 2024), ECoT (Zawalski et al., 2024), AHA (Duan et al., 2024), and OneTwoVLA (Lin et al., 2025).

**Benchmarks.** We conducted experiments on two simulation benchmarks, RLbench (James et al., 2020) and LIBERO-LONG (Liu et al., 2023); for real-world experiments, we utilized an Agilex Piper robotic arm.

### 4.2 Overall Performance

As shown in Table 1, Sentinel-VLA significantly outperforms all baselines consistently. Detailed analyses across different settings are provided below.

**Performance on RLBench Seen Tasks.** We first evaluated the model on 9 tasks from RLBench used during pre-training. Sentinel-VLA achieves a dominant average success rate of **63.5%**, significantly surpassing the base model PI0 (57.8%) and the CoT-based method ECoT (42.4%).

**Performance on RLBench Unseen Tasks.** To assess generalization, we evaluated the model on tasks that were not seen during pre-training. Sentinel-VLA maintains supe-

*Table 2.* Real-world average single action computation time of Sentinel-VLA and other comparative methods on an RTX4090.

| Method | OpenVLA | PI0 | ECoT | AHA+OpenVLA | OneTwoVLA | **Sentinel-VLA (Ours)** |
|---|---|---|---|---|---|---|
| **Computation time / action** | 57 ms | 8.5 ms | 1528 ms | 547 ms | 37ms | 13 ms |

*Table 3.* Real-world ablation study on SECL and OC-Adapter. SECL and OC-Adapter jointly enable the continual evolution.

| Variant | Stack | Pour | Sweep | **Avg** |
|---|---|---|---|---|
| Sentinel-VLA *w/o* SECL | 50% | 58% | 54% | 54.0% |
| *w/* SECL *w/o* OC-Adapter | 42% | 48% | 46% | 44.7% |
| **Sentinel-VLA (Full)** | **54%** | **66%** | **60%** | **60.0%** |

*Table 4.* Error detection rate of the Status Monitor on the RLBench evaluation set and the Real-world error status set.

| Evaluation Dataset | Detection Rate |
|---|---|
| RLBench Eval Set (Simulation) | 97.4% |
| Real-world Error Set (Real) | 90.6% |

*Table 5.* F1-Score of the Status Monitor Expert ($E_{sm}$) in simulation and real-world.

| Metric | Simulation | Real-World |
|---|---|---|
| Dataset Size | 100,000 transitions | 1,000 frames |
| Annotation | EC-Gen Ground Truth | 4 Human Experts |
| **F1-Score** | 0.9024 | 0.8567 |

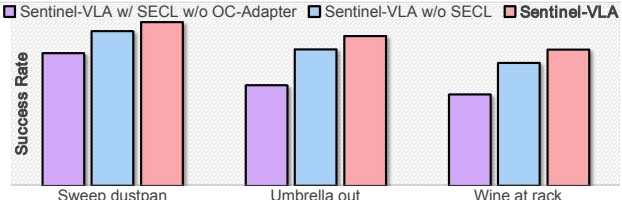

*Figure 5.* Ablation study of SECL and OC-Adapter on RLBench. They jointly enable the continual evolution.

rior performance with an average success rate of **51.3%**, outperforming PI0 (42.0%) and OpenVLA (30.7%) by a large margin. Specifically, in the challenging "Wine at rack" task which requires precise manipulation, Sentinel-VLA achieves 28%, triple the success rate of OpenVLA (8%). This indicates that the model is not merely memorizing trajectories but has learned generalized error recovery and planning capabilities that effectively transfer to novel task semantics and object interactions.

**Performance on LIBERO-LONG.** We further tested the model's long-horizon generalization on the LIBERO-LONG benchmark (Liu et al., 2023). Sentinel-VLA achieves a remarkable **90.7%** success rate. This represents a **5.5%** and **37.0%** improvement over PI0 (85.2%) and Open-VLA (53.7%), respectively. Despite the lack of domain-specific pre-training, our metacognitive architecture facilitates strong transfer learning, proving its robustness in handling long-horizon sequential decision-making where error accumulation typically leads to failure in baseline models.

**Robustness Under High Disturbance.** To test the effectiveness of our model under instability, we evaluated Sentinel-VLA in an RLBench setting with artificial disturbances (injecting 5% random perturbation probability per step, with a magnitude range of -0.1 to 0.1). As shown in the "Disturbed" section of Table 1, OpenVLA's performance collapses drastically from 35.6% to 25.6% as it lacks a mechanism to recognize that its trajectory has been perturbed. In contrast, Sentinel-VLA demonstrates exceptional resilience, maintaining a **54.7%** success rate.

**Performance on Real-world Tasks.** On a real-world Piper robot arm, Sentinel-VLA achieved an average success rate of **60.0%** across 3 tasks, outperforming PI0 (46.0%) and OpenVLA (30.7%) by nearly **25-30%**. As shown in Figure 6, the integrated error recovery was particularly crucial; for instance, in "Pour Water", the model recovered from

grasp failure, whereas baseline models simply halted or executed empty grasps. This result highlights the high practical utility of our approach in diverse physical embodiments.

**Inference Efficiency and Control Frequency.** A key advantage of Sentinel-VLA is its on-demand reasoning. To verify this, we tested the real-world average computation time against baselines on an RTX 4090. The results in Table 2 show that Sentinel-VLA operates at 13ms per action. This is significantly faster than CoT-based methods like ECoT (1528ms) and comparable to non-reasoning base models, validating that our model achieves high-level cognition without compromising real-time control frequency.

### 4.3 Component Analysis and Ablation Studies

**Performance of Status Monitor.** We evaluated the standalone performance of the Status Monitor Expert ($E_{sm}$) using 100k simulation transitions. Additionally, during real-world deployment, we collected 1,000 pairs of scenes and predicted statuses, which were manually judged by four human experts. To prevent the dominant "Normal" state from inflating metrics, we report the average F1-score across the critical *Initial*, *New-subtask*, and *Error* states. As shown in Table 5, $E_{sm}$ achieves high F1-scores in both simulation (90.2%) and the real world (85.7%), providing a reliable foundation for downstream dynamic reasoning.

**Generalization of Error Recognition.** To demonstrate that the status monitor can generalize to complex real-world errors, we annotated a separate real-world dataset of 1,000 error instances. We evaluated the model fine-tuned on real-world data against this set. As shown in Table 4, the model achieves a **90.6%** detection rate for diverse failure modes (e.g., gripper slip, wrong target object), which is comparable to its simulation performance of **97.4%**. This proves that

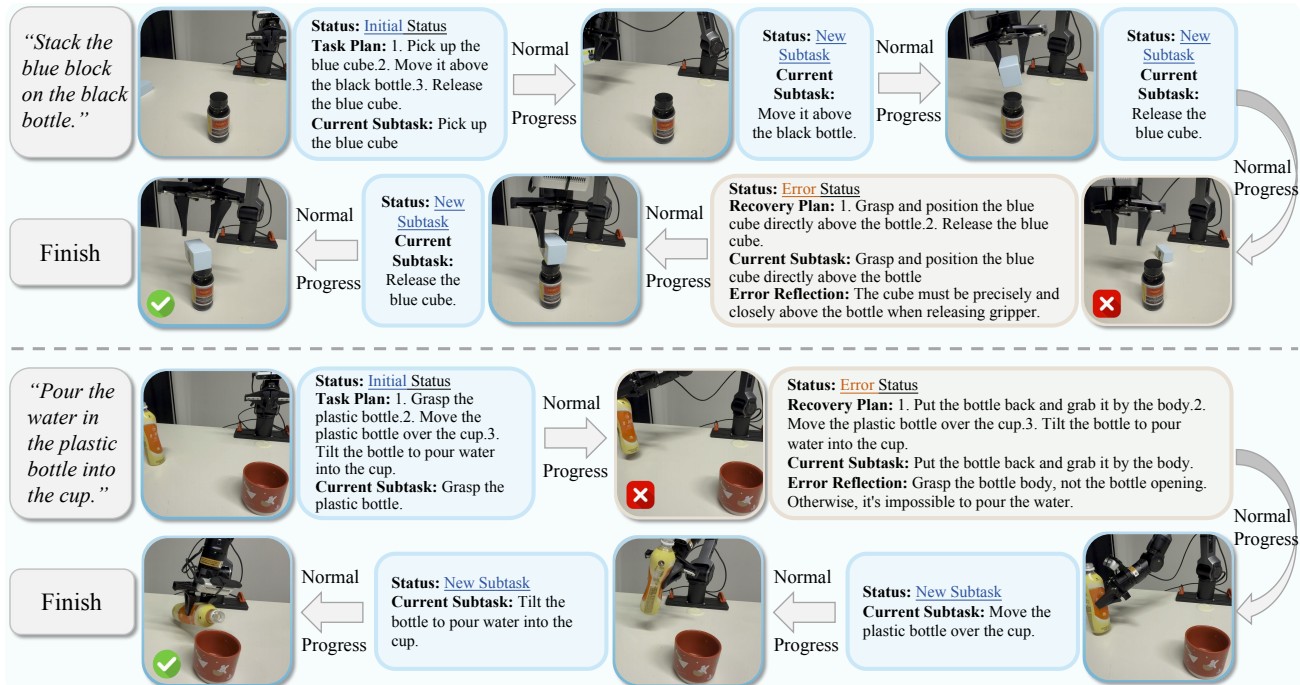

*Figure 6.* Two representative cases of Sentinel-VLA. It accomplished planning and recovery using only 4-5 reasoning times.

*Table 6.* Ablation study on model architecture across RLBench seen tasks. We compare the performance of OpenVLA trained with EC-Gen data, Sentinel-VLA without the status monitor (w/o SM), and the full Sentinel-VLA model.

| Task | OpenVLA (w/ EC-Gen) | Sentinel-VLA (w/o SM) | Sentinel-VLA (Full) |
|---|---|---|---|
| Close box | 74% | 90% | **94%** |
| Close laptop lid | 44% | 72% | **78%** |
| Toilet seat down | 80% | 100% | **100%** |
| Close fridge | 82% | 90% | **94%** |
| Frame off hanger | 24% | 74% | **74%** |
| Water plants | 20% | 44% | **46%** |
| Phone on base | 28% | 32% | **40%** |
| Put rubbish in bin | 12% | 22% | **24%** |
| Change clock | 18% | 22% | **22%** |
| **Average** | 42.4% | 60.7% | **62.6%** |

our monitor, despite being trained on synthetic data, effectively transfers its anomaly detection capabilities to visually distinct real-world failure scenarios.

**Architecture and Data Ablation.** To validate the effectiveness of our multi-expert architecture and the EC-Gen dataset, we benchmarked the full Sentinel-VLA against two variants: an OpenVLA trained on our EC-Gen data, and a Sentinel-VLA variant without the dedicated Status Monitor (w/o SM) (Table 6). In the w/o SM variant, the VLM expert directly generates the status token and the full reasoning chain autoregressively. As shown in the results, OpenVLA achieves a success rate of **42.4%** using our data, proving the intrinsic value of the synthesized error-recovery trajectories. However, removing the Status Monitor ($E_{sm}$)

causes Sentinel-VLA's performance to drop to **60.7%**. This degradation highlights a critical architectural insight: status prediction is inherently a discrete, option-based classification task. Assigning this task to a specialized, independent classifier ($E_{sm}$) is structurally more effective and reasonable than forcing the VLM expert to predict it as part of a continuous natural language generation process.

**Ablation on SECL and OC-Adapter.** Finally, to validate our continual learning strategy, we compared Sentinel-VLA under three conditions: without SECL, with SECL + LoRA, and with the full SECL + OC-Adapter. The real-world results are presented in Table 3, and RLBench results are in Figure 5. The model without SECL achieves **54.0%** in the real world. Simply applying LoRA for SECL leads to catastrophic forgetting, dropping the average success rate to **44.7%** (a -9.3% regression). In contrast, incorporating our SECL with OC-Adapter allows the model to evolve and achieve the highest success rate of **60.0%**. This confirms that the orthogonality constraint is essential for SECL.

## 5 Conclusion

We introduced ***Sentinel-VLA***, a unified cognitive VLA model that, for the first time, integrates dynamic deep thinking, status monitoring, and error recovery. This model's training was powered by our scalable EC-Gen pipeline, which automatically synthesized a large-scale dataset spanning 44 tasks and over 2.6 million transitions with rich annotations. Sentinel-VLA integrates a status monitor for efficient on-demand reasoning, ensuring robust decision-making with low overhead, while our SECL algorithm en-

ables continual learning from experience. Extensive experiments demonstrate Sentinel-VLA's superior performance over SOTA baselines, showcasing exceptional robustness and strong generalization, representing a significant step toward creating more robust and adaptive embodied agents.

## Impact Statement

This paper presents work whose goal is to advance the field of Machine Learning, specifically in the domain of Embodied AI and VLA models. The proposed methods for active status monitoring and dynamic error recovery have broad applications in automation, service robotics, and complex manipulation tasks. While this technology enables higher reliability and safety in physical environments, it also shares the general societal implications associated with advanced automation and robotics. There are no specific ethical concerns or adverse societal consequences unique to this work that we feel must be highlighted here.

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

# A    Details of EC-Gen

## A.1    Tasks used by EC-Gen

We used a total of 44 tasks in the EC-Gen data generation process, including: "beat_the_buzz", "change_channel", "change_clock", "close_box", "close_door", "close_fridge", "close_grill", "close_jar", "close_laptop_lid", "get_ice_from_fridge", "hang_frame_on_hanger", "hit_ball_with_queue", "insert_onto_square_peg", "insert_usb_in_computer", "open_door", "open_microwave", "open_washing_machine", "open_window", "open_wine_bottle", "phone_on_base", "pick_and_lift_small", "place_shape_in_shape_sorter", "play_jenga", "pour_from_cup_to_cup", "put_bottle_in_fridge", "put_groceries_in_cupboard", "put_item_in_drawer", "put_knife_in_knife_block", "put_knife_on_chopping_board", "put_money_in_safe", "put_rubbish_in_bin", "scoop_with_spatula", "setup_chess", "solve_puzzle", "stack_cups", "take_lid_off_saucepan", "take_off_weighing_scales", "take_plate_off_colored_dish_rack", "take_toilet_roll_off_stand", "take_tray_out_of_oven", "toilet_seat_down", "turn_oven_on", "tv_on", "water_plants".

## A.2    Status assignment

When annotating the status for trajectories generated by EC-Gen, we label the first 5 steps of each trajectory as **Initial Status**. The first 5 steps at the beginning of each non-erroneous subtask are labeled as **New-subtask Status**. The last 10 steps of the execution subsequence corresponding to the injected error waypoint are labeled as **Error Status**. All other steps are labeled as **Normal Status**.

## A.3    Error Reflection Generation

When generating error reflections for the errors we constructed, we use rule-based and template-based methods to generate the reflection text based on the error's type and specific content. For example, when we construct an *Object Interaction Error* where we suppress a "release gripper" operation that should have occurred at the error waypoint, we read the current interactive object "[object]" and the current subtask "[subtask]" from RLBench. One of the templates for this scenario is: "When executing [subtask], the gripper should be released when the [object] reaches the appropriate position, instead of continuing to move while holding it." We have multiple templates for each error category to ensure the diversity of the training data.

