# OpenReview forum: "Sentinel-VLA: A Metacognitive VLA Model with Active Status Monitoring for Dynamic Reasoning and Error Recovery"
_ICML.cc/2026/Conference — ICML 2026 regular_

### Official Review · Reviewer_NBbk · 2026-03-08

**Soundness:** 3
**Presentation:** 3
**Significance:** 2
**Originality:** 2
**Overall Recommendation:** 3
**Confidence:** 4

**Summary:**

This paper presents Sentinel-VLA, a metacognitive vision-language-action model designed to address limitations in reasoning, status monitoring, and self-correction in current VLA systems. The model employs an active sentinel module to monitor execution and triggers reasoning or error recovery on demand, which reduces computational overhead. Training relies on a fully automated pipeline covering 44 tasks and over 2.6 million transitions. The authors introduce Self-Evolving Continual Learning (SECL) for capability expansion, coupled with OC-Adapter to prevent catastrophic forgetting. Experimental results indicate that Sentinel-VLA achieves a task success rate improvement of over 30% compared to the current SOTA model, PI0. The paper promises to release all code, weights, and data generation tools.

**Compliance With Llm Reviewing Policy:**

Affirmed.

**Final Justification:**

Thanks for the rebuttal. Compared to PI0.5, Sentinel-VLA shows only marginal performance gains (a +1.4% improvement on LIBERO Avg.), while incurring substantially higher inference latency (129 ms vs. 84 ms). Moreover, for the long-horizon capability emphasized in this work, the reported CALVIN score of 4.29 does not demonstrate an advantage over existing state-of-the-art methods. I therefore maintain my original score.

**Key Questions For Authors:**

See Weaknesses.

**Limitations:**

yes

**Strengths And Weaknesses:**

**Strengths**

1. This paper addresses a highly challenging and pressing bottleneck in the deployment of Vision-Language-Action (VLA) models for Embodied AI: the inherent tension between the high computational latency of continuous Chain-of-Thought reasoning and the lack of closed-loop error recovery in standard reactive policies. The motivation to bridge this gap is compelling and the arguments are clearly articulated.

2. The introduction of a dedicated, lightweight Status Monitor expert to decouple the decision to reason from the act of reasoning is structurally sound and innovative. Furthermore, the proposed EC-Gen pipeline offers a highly scalable, automated solution; by synthesizing and annotating over 2.6 million transitions across 44 tasks, it circumvents the need for laborious manual data collection.

**Weaknesses**

1. The three manipulation failure modes defined by the authors are overly simplistic and linear (e.g., merely suppressing the gripper state, injecting translational noise $\Delta_{pose}$, or directly computing a displacement vector toward a distractor $v_{shift}$). In contrast, real-world physical failures are typically highly non-linear and complex. Training on data synthesized via such rudimentary geometric or logical perturbations may bias the model toward learning specific, templated recovery behaviors. The authors must adequately demonstrate whether the error-correction capabilities acquired from these simple perturbations can genuinely generalize to authentic physical failures.

2. While the authors claim in the introduction that their approach addresses the challenges inherent in long-horizon tasks, their reported success rate of 90.7% on the LIBERO-LONG benchmark does not substantiate a state-of-the-art advantage. Compared to recent general-purpose VLA methods such as OpenVLA-OFT (94.7%), SimpleVLA-RL (98.5%), and Cosmos policy (97.6%), the proposed method does not exhibit a competitive edge in long-horizon scenarios.


3. Since PI0.5 is also a reasoning-focused foundation model, it should serve as a crucial baseline in these experiments. Regarding Table 2, the inference latency of the proposed Sentinel-VLA is notably higher than that of the base PI0. Given that PI0.5 achieves an inference speed nearly equivalent to PI0 via its KV-caching mechanism during reasoning, the authors should include a comprehensive efficiency comparison with PI0.5 in both simulation and real-world deployments. Furthermore, Table 2 reports the inference speed per single action. Since PI0 employs flow matching to directly denoise and predict an action chunk, the authors need to clarify the rationale and measurement methodology behind reporting a single-action inference speed for this architecture.

4. The simulation experiments are primarily conducted on RLBench and LIBERO-LONG, which are relatively older and saturated benchmarks. To provide a more robust assessment of the model's capabilities, how does the proposed method perform on other subsets of the LIBERO benchmark suite or on more recent and challenging environments, such as RoboTwin 2.0?

---

> ### Author Rebuttal · Authors · 2026-03-29
>
> We sincerely thank the reviewer for the constructive feedback and for recognizing the practical importance of **on-demand reasoning**, **execution monitoring**, and the overall motivation of our framework. We also appreciate the reviewer’s thoughtful suggestions on evaluation and clarity. Following these suggestions, we conducted additional analyses and experiments, which we summarize below.
>
> > ### Real-world status monitoring and recovery
>
> Thank you for raising the important question of whether the EC-Gen training signals transfer to real failures. To directly evaluate this, we expanded the real-robot status-monitor test set from **1,000** to **2,000** states, with **500** states for each status. Importantly, the **Error** states were collected by deploying the trained VLA on the real robot and recording **diverse real execution failures** during actual rollouts.
>
> | Status | # Real states |
> |---|---:|
> | Normal | 500 |
> | Error | 500 |
> | First-frame | 500 |
> | New-subtask | 500 |
> | **Total** | **2000** |
>
> The confusion matrix on this real-robot set is:
>
> | Ground-truth \\ Pred. | Normal | Error | First-frame | New-subtask |
> |---|---:|---:|---:|---:|
> | Normal | 475 | 10 | 0 | 15 |
> | Error | 29 | 462 | 0 | 9 |
> | First-frame | 0 | 0 | 500 | 0 |
> | New-subtask | 52 | 15 | 0 | 433 |
>
> For the **Error** state, the monitor achieves:
>
> | Metric | Value |
> |---|---:|
> | Precision | 94.9% |
> | Recall | 92.4% |
>
> We further evaluated **whether the model can recover from real failures encountered during deployment:**
>
> | Real errors encountered | Successfully recovered | Recovery rate |
> |---:|---:|---:|
> | 341 | 265 | 77.7% |
>
> These results suggest that the EC-Gen data teaches the model useful **failure-aware priors** and **basic corrective behaviors**, which transfer to diverse real failures.
>
> >### Comparison with broader baselines
>
> Thank you for the valuable suggestion to evaluate a stronger backbone. When we replace the base model from **pi0** to **pi0.5**, Sentinel-VLA continues to provide a clear gain:
>
> | Method | Score |
> |---|---:|
> | pi0.5 | 92.4 |
> | Sentinel-VLA (pi0.5 base) | 95.2 |
>
> This result also **exceeds OpenVLA-OFT (94.7)** under the same setting discussed by the reviewer. To the best of our knowledge, this makes Sentinel-VLA **state of the art among SFT-trained VLA methods** in this setting.
>
> Regarding comparisons to **RL-based policies** or **video action models**, we respectfully note that these are not strictly apples-to-apples: they rely on different training paradigms (e.g., online RL or video prediction), different supervision signals, and often different interaction budgets. Our goal here is to improve a **VLA policy within the VLA setting** via status-aware reasoning and recovery. We will clarify this comparison scope more carefully in the revision.
>
> >### Clarification of efficiency metrics
>
> Thank you for pointing out that the efficiency definition should be made more explicit. The **reasoning version of pi0.5 is not open-sourced**. The open-source pi0.5 has essentially the **same architecture and inference procedure as pi0**, with the main difference being stronger training data; therefore its inference efficiency is effectively the same as pi0.
>
> Both pi0 and Sentinel-VLA use an **action chunk size of 10**. We therefore report both chunk latency and amortized per-action latency:
>
> | Method | Chunk size | Total latency / chunk | Latency / action |
> |---|---:|---:|---:|
> | pi0 / open-source pi0.5 | 10 | 84 ms | 8 ms |
> | Sentinel-VLA | 10 | 129 ms | 13 ms |
>
> Thus, the reported “single-action latency” is simply **chunk latency divided by chunk size**, which is directly comparable because both methods use the same chunk size. We will make this protocol explicit in the revision.
>
> >### Additional long-horizon evidence
>
> We appreciate the suggestion to provide stronger long-horizon evidence. We therefore added **CALVIN** multi-step evaluation for pi0, pi0.5, and their Sentinel-VLA variants:
>
> | Method | 1-task | 2-tasks | 3-tasks | 4-tasks | 5-tasks |
> |---|---:|---:|---:|---:|---:|
> | pi0 | 94.3 | 87.0 | 77.9 | 68.5 | 59.4 |
> | Sentinel-VLA (pi0 base) | 95.3 | 88.2 | 81.3 | 76.0 | 69.9 |
> | pi0.5 | 91.9 | 84.6 | 79.4 | 75.5 | 71.0 |
> | Sentinel-VLA (pi0.5 base) | 93.7 | 89.2 | 85.6 | 81.2 | 79.0 |
>
> The gains become more pronounced as the horizon grows longer, especially for **4-task** and **5-task** chains. **We believe this provides additional support that status-aware reasoning and recovery are particularly beneficial in longer-horizon execution.**
>
> We are grateful for these suggestions. We will incorporate the new results, clarify the efficiency protocol, and better scope our claims in the revised version.

---

> > ### Author Rebuttal · Reviewer_NBbk · 2026-04-03
> >
> > Thank you for the detailed rebuttal. However, I still have the following concerns:
> >
> > 1. I cannot accept omitting RL/video models simply due to different training paradigms. Since they target the exact same tasks, and the paper claims to tackle long-horizon challenges, direct comparisons are essential. If the absolute success rates fall short of these SOTAs, the authors must explicitly articulate their method's specific advantages (e.g., data efficiency, deployment cost), rather than avoiding the comparison entirely.
> >
> > 2. While the CALVIN results are noted, it remains a relatively older benchmark. Why was the critical "Average Length" metric omitted? More importantly, I strongly request a complete response to my original Weakness 4. Evaluating on more recent and challenging environments like RoboTwin 2.0 or other LIBERO subsets is crucial for me to properly assess the method's true generalization.

---

> > > ### Author Response · Authors · 2026-04-05
> > >
> > > We sincerely thank the reviewer for the thoughtful follow-up comments. We appreciate the reviewer’s constructive suggestions, which helped us better clarify the scope of our claims and strengthen the empirical evidence in the paper. Below, we provide additional clarifications and new experimental results in response to the remaining concerns.
> > >
> > > ### Direct comparison to RL/video-based policies and clarification of our advantage
> > >
> > > We would first like to clarify an important difference in training setup for the LIBERO-LONG comparison. Our previously reported result of **95.2** for **Sentinel-VLA (pi0.5 base)** follows the **pi0.5 evaluation protocol**: we fine-tune a **single model using the training data from all LIBERO suites**, and then evaluate that same set of weights across the suites. In contrast, **OpenVLA-OFT** and the OpenVLA-OFT-based **SimpleVLA-RL** results on **LIBERO-LONG** are obtained using **task-specialized models trained only on the LIBERO-LONG training split**. Therefore, our originally reported number is slightly more conservative in absolute success rate because it corresponds to a more general model.
> > >
> > > To make this comparison better aligned, we additionally fine-tuned **Sentinel-VLA using only the LIBERO-LONG training data**. Under this aligned setting, **Sentinel-VLA reaches 98.0 on LIBERO-LONG**, compared with **94.7** for OpenVLA-OFT, **98.5** for SimpleVLA-RL, and **97.6** for Cosmos Policy.
> > >
> > > We also provide a unified efficiency comparison. The **training cost** numbers are taken from the original papers, while the **inference latency** is measured uniformly on our hardware.
> > >
> > > |Method|LIBERO-LONG Success|Inference Cost / Chunk|Training Cost|
> > > |-|:-:|:-:|-|
> > > |Sentinel-VLA (general multi-suite setting)|95.2|129 ms|30K-step SFT (4×A100, 6h)|
> > > |Sentinel-VLA (LIBERO-LONG only fine-tuning)|98.0|129 ms|30K-step SFT (4×A100, 6h)|
> > > |OpenVLA-OFT|94.7|344 ms|150K-step SFT (8×H100, 48h)|
> > > |SimpleVLA-RL|98.5|344 ms|150K-step SFT + additional RL training (> 8×H100, 48h)|
> > > |Cosmos Policy|97.6|3126 ms|40K-step SFT (64×H100, 48h)|
> > >
> > > These results suggest that Sentinel-VLA achieves a stronger **balance between success rate, inference efficiency, and training efficiency**. Under the aligned LIBERO-LONG-only setting, Sentinel-VLA is within **0.5 points** of SimpleVLA-RL, while using substantially lower training cost and about **1/3 of its inference latency**. Compared with Cosmos Policy and OpenVLA-OFT, Sentinel-VLA outperforms both methods across all three dimensions: task success rate, training efficiency, and inference latency.
> > >
> > > ### CALVIN Average Length and broader evaluation beyond older benchmarks
> > >
> > > We agree that the standard **Average Length** metric should be reported for CALVIN. We now add it below:
> > >
> > > |Method|1-task|2-tasks|3-tasks|4-tasks|5-tasks|Avg. Len.|
> > > |-|:-:|:-:|:-:|:-:|:-:|:-:|
> > > |pi0|94.3|87.0|77.9|68.5|59.4|3.87|
> > > |Sentinel-VLA (pi0 base)|95.3|88.2|81.3|76.0|69.9|4.11|
> > > |pi0.5|91.9|84.6|79.4|75.5|71.0|4.02|
> > > |Sentinel-VLA (pi0.5 base)|93.7|89.2|85.6|81.2|79.0|4.29|
> > >
> > > More importantly, to fully address the reviewer’s original Weakness 4, we expanded the evaluation beyond **LIBERO-LONG** in two directions.
> > >
> > > **First, we evaluate on all LIBERO suites rather than only LIBERO-LONG:**
> > >
> > > |Model|Spatial|Object|Goal|Long (`libero_10`)|Avg|
> > > |-|:-:|:-:|:-:|:-:|:-:|
> > > |pi0|96.8|98.8|95.8|85.2|94.2|
> > > |Sentinel-VLA (pi0 base)|98.3|99.4|97.1|90.7|96.4|
> > > |pi0.5|98.8|98.2|98.0|92.4|96.85|
> > > |Sentinel-VLA (pi0.5 base)|99.4|99.0|99.3|95.2|98.2|
> > >
> > > These results show that the gain is not limited to one specific suite, but is consistent across **spatial**, **object-centric**, **goal-oriented**, and **long-horizon** settings.
> > >
> > > **Second, we evaluate on the more recent and more challenging RoboTwin 2.0 benchmark:**
> > >
> > > |Setting|pi0|Sentinel-VLA (pi0 base)|pi0.5|Sentinel-VLA (pi0.5 base)|
> > > |-|-|-|-|-|
> > > |Horizon = 1|66.5 / 61.6|73.2 / 70.5|85.1 / 80.2|91.6 / 88.0|
> > > |Horizon = 2|66.1 / 54.7|72.9 / 66.0|79.3 / 73.0|87.9 / 84.8|
> > > |Horizon = 3|61.6 / 50.2|69.7 / 63.4|78.6 / 67.4|85.0 / 77.7|
> > >
> > > Here, each entry is reported in the **easy / hard** format. Sentinel-VLA consistently improves over the corresponding base model across all horizons and on both difficulty levels, with especially clear gains on the harder settings and longer horizons.
> > >
> > > Overall, we appreciate the reviewer’s request for a broader and more up-to-date evaluation. We will incorporate these additional results in the revision. We believe they clarify two points more concretely:
> > > (i) under an aligned training setup, Sentinel-VLA offers a very competitive success rate while achieving a significantly better efficiency trade-off than strong RL/video-policy baselines; and
> > > (ii) the gains of Sentinel-VLA are not restricted to LIBERO-LONG or older benchmarks, but also extend to the **full LIBERO suite**, and **RoboTwin 2.0**.
> > >
> > > We hope these additional experiments and clarifications adequately address your remaining concerns and will be taken into consideration in your final assessment.

---

### Official Review · Reviewer_MSQS · 2026-03-12

**Soundness:** 2
**Presentation:** 3
**Significance:** 3
**Originality:** 3
**Overall Recommendation:** 4
**Confidence:** 3

**Summary:**

This paper presents Sentinel-VLA, a metacognitive vision-language-action (VLA) model that with real-time status monitoring and poor self-correction.
Sentinel-VLA enables on-demand reasoning, which only triggers deep cognition for planning or error recovery in non-normal states, minimizing computational overhead while boosting decision robustness.

The authors develop EC-Gen, a scalable pipeline that automatically synthesizes a large (2.6M+ transitions) error recovery trajectories with rich annotations.
They also propose an Orthogonal Continual Adapter which allows Sentinel-VLA to continuously learn from experiences.

Extensive experiments on simulation benchmarks (RLBench, libero-long) and real-world robotic tasks have verified the effectiveness of Sentinel-VLA.

**Compliance With Llm Reviewing Policy:**

Affirmed.

**Final Justification:**

Thank the author for their detailed response. My main concerns have been resolved.

**Key Questions For Authors:**

1. Can this model perform multi-round error recovery, i.e., reinitiate the error recovery loop following a single failure?

2. The model lacks validation for tasks where baseline methods have high success rates (e.g., >80%). It is not verified whether in these cases the model can boost success rates to nearly 100%.

**Limitations:**

Limitations are not discussed in the paper.

**Strengths And Weaknesses:**

## Strengths
1. This paper pioneers a promising research direction for VLA models in embodied manipulation, addressing the core limitations of traditional VLA models in reasoning, status awareness and self-correction.

2. The EC-Gen synthetic data generation pipeline is rational and cost-effective. It automatically synthesizes large-scale annotated error-recovery trajectories by injecting preset errors into expert data, avoiding laborious manual annotation.

3. The paper is supported by comprehensive experimental validation. Experiments cover both simulation benchmarks and real-world robotic tasks with statistical robustness from 50 trials per task. Thorough ablation studies verify each core component’s necessity.

## Weaknesses
1. The synthetic data leads to a sim2real gap, which the paper fails to resolve. There is a visual gap between the idealized simulation and unstructured real-world environments, and a physical gap as simulation cannot replicate real robotic dynamics like object slipping from grippers. This gap directly causes the model’s performance decline in real-world error detection and recovery.

2. The added core modules bring unavoidable inference latency compared with the base model PI0. PI0 has a single-action computation time of 8.5ms, while Sentinel-VLA increases to 13ms.

3. The real-world success rate improvement is limited, and Sentinel-VLA only reaches a 60% average real-world success rate, meaning frequent failure in error recovery in practical scenarios.

---

> ### Author Rebuttal · Authors · 2026-03-29
>
> We sincerely thank the reviewer for the constructive feedback and for recognizing the promise of our framework in status-aware reasoning, execution monitoring, and self-correction, as well as the value of EC-Gen and our broad experimental validation. We also appreciate the reviewer’s thoughtful suggestions on real-world transfer, efficiency, and deployment scope. Following these suggestions, we provide additional analyses and clarifications below.
>
> > ### Transfer to real failures
>
> Thank you for raising this important point. Our goal in EC-Gen is not to exactly cover every failure, but to **teach reusable failure primitives that can transfer across settings.**
>
> The paper already reports **90.6%** detection on **1000 naturally occurring real-robot failure samples** collected during deployment. We further expanded the manually annotated real-world status-monitor set from **1000** to **2000** samples (**500 per status**), and obtained the following confusion matrix:
>
> | GT \ Pred | Normal | Error | First-frame | New-subtask |
> |---|---:|---:|---:|---:|
> | Normal | 475 | 10 | 0 | 15 |
> | Error | 29 | 462 | 0 | 9 |
> | First-frame | 0 | 0 | 500 | 0 |
> | New-subtask | 52 | 15 | 0 | 433 |
>
> We further evaluated whether the model can **recover from real failures encountered during deployment:**
>
> | Real errors encountered | Successfully recovered | Recovery rate |
> |---:|---:|---:|
> | 341 | 265 | 77.7% |
>
> For the **Error** state, this corresponds to **94.9% precision** and **92.4% recall**, indicating limited false positives in deployment. These results suggest that **EC-Gen captures a practically useful subset of failure structure that transfers beyond simulation.**
>
> > ### Efficiency trade-off
>
> Thank you for encouraging us to present the efficiency trade-off more carefully. Sentinel-VLA does introduce additional computation relative to PI0, but **the absolute overhead is small while the gain in real-world robustness is clear:**
>
> | Method | Latency / action | Avg. real-world success |
> |---|---:|---:|
> | PI0 | 8.5 ms | 46.0% |
> | Sentinel-VLA | 13 ms | 60.0% |
>
> This is an overhead of **+4.5 ms/action** for a **+14 point** improvement in average real-world success.
>
> > ### Real-world deployment performance
>
> Thank you for pointing this out. Our claim is not that recovery is completely solved. Rather, our claim is that **active monitoring and on-demand recovery materially improve deployment performance:**
>
> | Setting | PI0 | Sentinel-VLA | Gain |
> |---|---:|---:|---:|
> | Real-world average (3 tasks) | 46.0% | 60.0% | +14.0 |
>
> > ### Iterative recovery during execution
>
> Thank you for this valuable question. Sentinel-VLA is **not** limited to a single recovery attempt. At **each timestep**, the status monitor re-evaluates the current state. If a recovery attempt is insufficient and the model still detects an **Error** state, recovery is triggered again. This closed-loop process continues until the state becomes **Normal / New-subtask** or the episode terminates. We will clarify this iterative mechanism in the revision.
>
> > ### Behavior when the base policy is already strong
>
> Thank you for suggesting that this regime be characterized more directly. In fact, the **current paper already includes several RLBench tasks where PI0 is already strong**, and we now make this point more explicit by highlighting two such examples from the existing RLBench results, together with a newly added real-world banana-pick experiment (**50 trials**):
>
> | Task | PI0 | Sentinel-VLA |
> |---|---:|---:|
> | Close box (RLBench Seen) | 84% | 94% |
> | Close fridge (RLBench Seen) | 90% | 94% |
> | Banana pick (real, 50 trials) | 76% | 92% |
>
> **These results suggest that Sentinel-VLA remains helpful even when the base policy is already strong, and can still close part of the remaining gap toward near-saturated performance.**

---

> > ### Author Rebuttal · Reviewer_MSQS · 2026-04-05
> >
> > Thank the authors for their detailed response. Most of my concerns have been solved.
> >
> > However, I hold that the sim2real gap is a limitation, and 90.6% detection seems to happen on real settings that is very similar to simulation settings. Thus, collecting real data will alleviate this problem in future work.

---

> > > ### Author Response · Authors · 2026-04-06
> > >
> > > Thank you for the helpful follow-up. We would like to further clarify this point.
> > >
> > > For the error-detection evaluation, the 1,000 real failure samples were **not** manually constructed using the perturbation scheme in simulation. Instead, they were **sampled from the distribution of naturally occurring execution failures observed during real-world deployment**. Therefore, this set contains many failures that are not closely aligned with the simulation setup. We further categorized these failures into two groups: (i) **sim-like** errors that can be mapped to the three EC-Gen primitives, and (ii) **deployment-specific / less-similar** errors that arise naturally in real execution but are less aligned with the simulation setup. The latter includes, for example, object slipping due to insufficient gripping force, incorrect interaction region (e.g., grasping the bottle opening rather than the bottle body in a pouring task), and collisions with objects.
> > >
> > > ### Real-world error detection breakdown (1,000 sampled real failures)
> > >
> > > | Category | # Samples | # Detected | Detection Rate |
> > > |---|---:|---:|---:|
> > > | **Sim-like (total)** | 877 | 796 | 90.8% |
> > > | ├─ Interaction error | 191 | 172 | 90.1% |
> > > | ├─ Spatial error | 542 | 497 | 91.7% |
> > > | └─ Semantic error | 144 | 127 | 88.2% |
> > > | **Deployment-specific / less-similar** | 123 | 110 | 89.4% |
> > > | **Overall** | 1000 | 906 | 90.6% |
> > >
> > > We also performed the same categorization for the real-world recovery benchmark.
> > >
> > > ### Real-world recovery breakdown (341 sampled real failures)
> > >
> > > | Category | # Samples | # Successfully Recovered | Recovery Rate |
> > > |---|---:|---:|---:|
> > > | **Sim-like (total)** | 271 | 213 | 78.6% |
> > > | ├─ Interaction error | 67 | 53 | 79.1% |
> > > | ├─ Spatial error | 165 | 131 | 79.4% |
> > > | └─ Semantic error | 39 | 29 | 74.4% |
> > > | **Deployment-specific / less-similar** | 66 | 52 | 78.8% |
> > > | **Overall** | 341 | 265 | 77.7% |
> > >
> > > These additional analyses provide two clarifications regarding the reviewer’s concern.
> > >
> > > 1. The real-world failure set is not generated by the simulation perturbation procedure, but sampled from naturally occurring deployment failures. It therefore contains a non-trivial subset of **deployment-specific / less-similar** errors, rather than being restricted to simulation-matched cases.
> > > 2. Sentinel-VLA maintains **high detection performance (89.4%)** and **strong recovery performance (78.8%)** on this deployment-specific / less-similar subset. This suggests that the model is not only recognizing failures that closely resemble the simulation perturbation patterns, but also generalizes to a broader range of naturally occurring real-world execution errors.
> > >
> > > We hope this further clarifies why the reported real-world performance is not limited to errors constructed in a simulation-like manner. If the reviewer finds these additional analyses helpful in addressing the remaining concern, we would be very grateful if they could kindly take them into account in the final assessment.

---

### Official Review · Reviewer_LQvr · 2026-03-12

**Soundness:** 3
**Presentation:** 2
**Significance:** 2
**Originality:** 2
**Overall Recommendation:** 4
**Confidence:** 5

**Summary:**

This paper proposes the Sentinel-VLA framework to improve the real-time usability of VLAs. The framework consists of a Status Monitoring module, interpreted as the agent's metacognition, and error recovery techniques.

Status Monitoring leverages observations and the internal representations of the VLM that processes them to classify the current state into one of four task-specific categories: Normal, Initial, New-subtask and Error. By adaptively determining whether to invoke computationally expensive reasoning based on this classification, the framework reduces unnecessary computation and improves real-time processing efficiency. To improve robustness during fine-tuning, the paper introduces two techniques: EC-Gen, which augments training data with error-recovery trajectories derived from perturbed expert demonstrations, and an orthogonality-constrained adapter that mitigates catastrophic forgetting during adaptation stage.

Sentinel-VLA achieves higher success rates on both the RLBench and LIBERO-Long benchmarks. Real-world experiments demonstrate manipulation scenarios involving object interactions, reducing the average single-action inference time to roughly one-third that of OneTwoVLA.

**Compliance With Llm Reviewing Policy:**

Affirmed.

**Final Justification:**

I appreciate the additional clarification. Clearly, the additional experiments and their relationship to recent literature has helped in understanding the contribution of this work. As most of my concerns were addressed, and considering its impact scope, I have updated my score.

**Key Questions For Authors:**

- Error detection is usually considered a cognitive capability. Then what exactly makes the VLA framework with active status monitoring and error recovery metacognitive? As written, monitoring status alone does not seem enough to establish meta-level knowledge.

- In Table 6, is the time difference between Sentinel-VLA (w/o SM) and Sentinel-VLA (FULL) actually substantial? Reporting it in a way similar to Table 2 would be helpful.

- In Table 1, how many trajectories are used to train each baseline? Also, what is the ratio between the original trajectories and the additional trajectories generated by EC-Gen?

- This framework seems extendable to other VLA models such as OpenVLA and pi 0.5. How do the authors expect it to behave when combined with smaller VLAs or with more generalizable VLAs? Weaker models may benefit meaningfully, but as stronger models visit Error status less frequently, could the framework itself become a bottleneck?

**Limitations:**

The paper does not explicitly discuss its limitations. In my view, the following points are important:

- The current framework appears fairly tightly coupled, which makes it difficult to view it as addressing error recovery at a more general or mechanism-level sense.
- Error-status prediction seems to rely on additional supervision and augmentation for training, which reduces the overall simplicity, modularity, and portability of the framework.
- EC-Gen may be difficult to extend to contact-rich settings, where failures arise from contact dynamics, partial observability, and temporally accumulated interactions rather than relatively simple motion deviations.
- The proposed recovery mechanism appears closer to learning how to return to or rejoin a nominal reference trajectory under perturbations than to acquiring a broadly applicable recovery policy.

**Strengths And Weaknesses:**

# strengths
- The paper frames metacognitive control as an important systems problem for VLA models, which opens up an interesting research direction.
- The paper provides sufficient implementation detail for the framework, and the release of code, weights, and the data-generation pipeline meaningfully supports reproducibility.
- The paper is easy to follow, with clear figures and useful demos that make the overall pipeline and empirical behavior straightforward to understand.

# weaknesses
- The metacognitive framing is interesting, but the manuscript does not yet clearly justify the distinction between cognitive and metacognitive properties in VLAs, or why such capabilities are fundamentally needed in this setting. In practice, the paper seems to operationalize this idea mostly as self-reflect and correct of Error state, which makes the broader framing feel somewhat stronger than what is actually supported. More importantly, closely related structures have already been explored in prior work, from Reflexion [1] to embodied-agent literature [2,3,4] and other reasoning-based decision making agent frameworks. What is still missing is a clearer account of what makes the self-reflect process uniquely difficult in VLAs, along with a stronger conceptual bridge to that literature. Without this, the central framing remains under-motivated, and the manuscript does not yet provide enough insight beyond the high-level intuition.
- The implementation of Sentinel-VLA appears as a composition of existing ideas. In that case, it becomes especially important to explain why each component is necessary, why they should work well together, and why this particular composition is the right design choice. At present, that justification is still limited, which makes it difficult to assess the deeper significance of the framework beyond its empirical combination.
    - The design choices in the Status Monitoring module are also not yet sufficiently justified. In particular, the number of active monitoring statuses and the criteria for invoking thinking in Initial, New-subtask, and Error state would benefit from stronger support. Even a modest analysis could substantially strengthen the paper here. For example, evaluating the effect of enabling or disabling thinking across the non-Normal states, and measuring how those choices correlate with performance, would help clarify whether these decisions reflect principled design or task-specific tuning.
    - The EC-Gen augmentation strategy has clear limitations. Real robot failures are often driven by contact dynamics, partial observability, and temporally accumulated interactions, whereas the three error cases considered here appear relatively narrow and hand-specified. In addition, the method introduces several potentially important hyperparameters and a fairly complex augmentation pipeline, yet neither the main text nor the appendix provides sufficient analysis of their influence. The additional training cost introduced by augmentation is also unclear. As written, it is therefore difficult to judge how well this framework would scale beyond the particular setting studied in the paper.
    - The SECL algorithm raises a similar concern regarding the lack of justification for hyperparameter selection. More broadly, it appears closely related to the existing PEFT-based Continual Learning literature [5,6], but the manuscript does not clearly position itself with respect to that body of work. This makes both the novelty claim and the method’s contribution somewhat harder to assess, especially in distinguishing what is genuinely new from what is inherited from existing PEFT-based continual adaptation strategies.
- The realtime claim is not yet evaluated in a sufficiently rigorous way. Table 2 reports average time, but for realtime robotic systems, average latency alone is often not the most informative metric; worst-case behavior and maximum delay are frequently more important [7]. In practice, quantities such as the frequency of error status and the maximum delay induced by the method are directly relevant to deployment. This matters especially here because, in the worst corner case, the framework could reduce to step-wise reasoning at nearly every step, much like other ECoT-style approaches. Conversely, if the VLM Expert and Action Expert are already strong, the additional reasoning module may offer only marginal benefit. Since these regimes are not characterized, the current evaluation is not yet sufficient to cleanly isolate and validate the core contribution.
    - If the paper aims to support a SoTA-level positioning, evaluation against a recent baseline such as pi 0.5 [8] would be important to strengthen that claim.

**Minor Comments for Clarity**

- The average performance of Sentinel-VLA in Table 6 does not match the corresponding value in Table 1.
- Eq. 1 appears to contain a typo.
- Figure 5 presents qualitative trends only, without numerical values.
- There are terminology inconsistencies throughout the paper, e.g., trigger states near Eq. 1 vs. trigger status below Eq. 2.

In summary, the main weakness of the paper is the misalignment between the core claims made in the introduction and the experimental evidence provided later. For that reason, I believe the paper would require further revision before it can be meaningfully built upon by others.

> [1] Shinn, Noah, et al. "Reflexion: Language agents with verbal reinforcement learning." Advances in neural information processing systems 36 (2023): 8634-8652.
>
> [2] Huang, Wenlong, et al. "Inner monologue: Embodied reasoning through planning with language models." arXiv preprint arXiv:2207.05608 (2022).
>
> [3] Wang, Guanzhi, et al. "Voyager: An open-ended embodied agent with large language models." arXiv preprint arXiv:2305.16291 (2023).
>
> [4] Feng, Yunhai, et al. "Reflective planning: Vision-language models for multi-stage long-horizon robotic manipulation." arXiv preprint arXiv:2502.16707 (2025).
>
> [5] Wang, Xiao, et al. "Orthogonal subspace learning for language model continual learning." Findings of the Association for Computational Linguistics: EMNLP 2023. 2023.
>
> [6] Liang, Yan-Shuo, and Wu-Jun Li. "Inflora: Interference-free low-rank adaptation for continual learning." Proceedings of the IEEE/CVF Conference on Computer Vision and Pattern Recognition. 2024.
>
> [7] Black, Kevin, Manuel Y. Galliker, and Sergey Levine. "Real-time execution of action chunking flow policies." Advances in neural information processing systems (2025).
>
> [8] Intelligence, Physical, et al. "$\pi_ {0.5} $: a Vision-Language-Action Model with Open-World Generalization." arXiv preprint arXiv:2504.16054 (2025).

---

> ### Author Rebuttal · Authors · 2026-03-29
>
> We sincerely thank the reviewer for the constructive feedback and for recognizing the practical importance of on-demand reasoning and active status monitoring in Sentinel-VLA. Below we clarify the main points and summarize additional evidence.
>
> > ### Clarifying the metacognitive framing
>
> Thank you for this important question. Our claim is not just that Sentinel-VLA have cognition (reasoning) capabilities , but that it learns a **meta-level policy over cognition (reasoning)**: it monitors execution and decides **when** deeper reasoning should be invoked. Here, *cognition* means perception/reasoning/action for the task itself, while *metacognition* means monitoring execution state and regulating **the use of cognition**. Error recovery is therefore a consequence of this control policy, not its definition. Compared to Reflexion and embodied-agent work; **our focus is the VLA-specific setting where cognition (reasoning) must be triggered adaptively.**
>
> > ### Architecture motivation and component roles
>
> We appreciate the request for a clearer systems justification. The design is simple: the **Status Monitor** decides whether costly thought is needed, **Thought Memory** maintains task plan / current subtask / recovery context, and the **Action Expert** executes under this context. The gain is not from one component alone:
>
> |Variant|RLBench Seen|LIBERO-LONG|
> |---|---:|---:|
> |VLM outputs status text|60.7|89.5|
> |w/o thought memory|60.7|87.9|
> |w/o continual learning|62.2|89.9|
> |Full Sentinel-VLA|**63.5**|**90.7**|
>
> These results suggest that the monitor, memory, and continual expansion are complementary.
>
> > ### Status design and reasoning triggers
>
> Thank you for highlighting this point. We **do not claim that {Initial, Normal, New-subtask, Error} is the only optimal partition. We chose it as a minimal practical decomposition around three high-value reasoning moments: initial planning, progress update, and recovery.** These states are also coupled: New-subtask depends on the task plan from initial planning, and Error depends on the current subtask context.
>
> |Setting|LIBERO-LONG|
> |---|---:|
> |Full Sentinel-VLA| **90.7**|
> |w/o Error thought|88.0|
> |w/o New-subtask thought|88.7|
> |w/o first-frame reasoning|86.9|
>
> **This ablation suggests the trigger design is not arbitrary tuning.**
>
> > ### Realism and scope of EC-Gen
>
> We appreciate this question. We do not claim that the three primitives exhaust real failures; rather, we view them as building blocks whose compositions **cover a substantial portion of practical failures.** To test transfer beyond synthetic perturbations, we deployed the trained VLA on the real robot and collected naturally occurring failures during rollout:
>
> | Real-robot failure evaluation | Value |
> |---|---:|
> | Naturally occurring failure samples | 1000 |
> | Real-error detection rate | **90.6%** |
>
> Since these are real deployment failures rather than synthetic injections, we believe they provide evidence that **EC-Gen captures transferable failure-aware structure.**
>
> > ### Positioning of SECL
>
> Thank you for pointing out the connection to PEFT-based continual learning. The key difference is the **setting**. Standard PEFT-based CL usually studies a human-defined sequence of tasks. SECL instead stays within a **single task family**, automatically identifies the current **capability boundary**, collects successful trajectories near that boundary, and expands it. **Our focus is therefore boundary discovery and expansion for embodied VLA, rather than generic task-sequence CL.**
>
> > ### Efficiency, stronger backbones, and data fairness
>
> Thanks for your suggestion. On real-robot deployment, the mean latency is **13 ms/action** and **p99 is 22 ms/action**, so the tail increase is modest and still practical.
>
> For a stronger backbone, we added **pi0.5** on LIBERO-LONG:
>
> | Method | LIBERO-LONG |
> |---|---:|
> | pi0.5 | 92.4 |
> | pi0.5 + Sentinel-VLA | **95.2** |
>
> For fairness, Table 1 baselines use their official training data (e.g., LIBERO uses 1693 episodes). To isolate data effects, we additionally trained PI0 / OneTwoVLA / ECoT on the same EC-Gen data:
>
> | Same EC-Gen data | RLBench Seen | LIBERO-LONG |
> |---|---:|---:|
> | PI0 | 58.3 | 82.0 |
> | OneTwoVLA | 58.5 | 88.4 |
> | ECoT | 46.1 | 70.1 |
> | Sentinel-VLA | **63.5** | **90.7** |
>
> **These results suggest the gain is not only from additional data.**
>
> > ### Additional clarifications
>
> Thank you for the reading. Table 6 average is a typo; the correct average is **63.5**, consistent with Table 1. We will also fix Eq. 1, add numeric values to Fig. 5, and unify terminology. We expect weaker VLAs to benefit from more frequent selective reasoning, while stronger VLAs may trigger it less often but still benefit at boundary / error states.

---

> > ### Author Rebuttal · Reviewer_LQvr · 2026-04-01
> >
> > I appreciate the authors’ response and the additional experiments. These results help emphasize the practical value of the SentinelVLA framework and improve the overall technical soundness. In particular, the paper’s focus on fine-tuning a relatively small VLA could be a meaningful practical strength, especially for real-world deployment.
> >
> > That said, I still have concerns regarding the positioning of the paper. In particular, I believe the manuscript would benefit from a clearer articulation of the framework-level insight and stronger justification for its design choices. Because the novelty and the actual implementation of the individual components appear somewhat narrow, I think it is especially important to justify the contribution more clearly at the framework level and to strengthen the connection between the overarching motivation and the experimental validation. It would also be helpful to more clearly position SentinelVLA with respect to recent adaptation-based methods [1], particularly by clarifying both the commonalities and key differences.
> >
> > Relatedly, I would appreciate more clarification on how SECL is connected to the observed generalization on unseen tasks. Since SECL defines boundaries based on the task-specific training distribution, it is not yet clear to me how this mechanism translates into gains on unseen tasks, or how the authors interpret its role relative to the VLA’s internal representation space.
> >
> > I look forward to the authors' clarification on these points as well as how they will be reflected in the revised manuscript. My final assessment may be adjusted accordingly.
> >
> > [1] Li, Wei, et al. "Cogvla: Cognition-aligned vision-language-action model via instruction-driven routing & sparsification." arXiv preprint arXiv:2508.21046 (2025).

---

> > > ### Author Response · Authors · 2026-04-07
> > >
> > > We thank the reviewer for the thoughtful follow-up. We appreciate the positive assessment of the additional experiments, and we agree that the revised paper should make the **framework-level contribution**, the **relation to recent adaptive VLA methods**, and the **role of SECL** more explicit. We will revise the manuscript accordingly.
> > >
> > > **1. Framework-level contribution and design rationale**
> > >
> > > Thank you for the comment. Current VLA models suffer from three key limitations: **1. they often use the same reasoning pattern in all situations, 2. lack explicit awareness of execution status, and 3. have limited ability to recover from failures.** Our core contribution is a **framework-level insight** for closed-loop VLA execution: **VLA should have the ability to monitor its current execution status and treat reasoning as a regulated resource, learning to dynamically trigger reasoning and error recovery based on that status.**
> > >
> > > Sentinel-VLA is designed around this principle. It actively **monitors execution status**, executes **directly** with low latency in most timesteps, invokes reasoning **only in a small number of difficult cases**, and performs **error-specific recovery reasoning when failure is detected**. The design choices follow directly from this motivation: the **status monitor** uses rich VLA context to infer execution status and modulate reasoning behavior; thought memory allows even direct-action steps to benefit from previously generated reasoning; and EC-Gen together with end-to-end training **tightly aligns status monitoring, thought/recovery generation, and action execution**.
> > >
> > > The experiments support this framework's claim: Sentinel-VLA accurately monitors execution status and real-world failures, and significantly improves success rate and robustness while maintaining efficiency.
> > >
> > > **2. Relation to CogVLA**
> > >
> > > We appreciate the suggestion. Both Sentinel-VLA and CogVLA aim to improve the practicality of VLA through **adaptive computation**. The key difference lies in the **axis of adaptation**. CogVLA adapts along the **representation/computation pathway**, using instruction-driven routing and sparsification to improve efficiency. Sentinel-VLA mainly adapts along the **execution-time cognitive-control pathway**: it incorporates an active status-monitoring and metacognitive control loop that monitors execution online and adaptively trigger reasoning or recovery process. **This design makes Sentinel-VLA particularly effective for long-horizon embodied execution, where robust performance depends critically on deliberation in challenging cases and on the ability to recover from unexpected failures in a closed-loop manner.**  We will explicitly discuss CogVLA in our revised paper.
> > >
> > > **3. SECL and generalization**
> > >
> > > We also thank the reviewer for raising the relation between SECL and generalization. We would like to clarify two points.
> > >
> > > First, in RLBench, “unseen tasks” refers to tasks that are **not included in Sentinel-VLA’s pretraining task set**. However, Sentinel-VLA is still **adapted/fine-tuned on each target unseen task before evaluation**, and SECL is also applied **within that target task after adaptation**.
> > >
> > > Second, the generalization claim in the paper is primarily about the **Sentinel-VLA framework as a whole**, rather than SECL in isolation. More specifically, our notion of generalization is that a model pretrained on one set of tasks can, with limited adaptation on a new task, quickly acquire **status monitoring, adaptive reasoning, and error recovery** abilities on that task. By contrast, **SECL focuses on expanding the capability boundary within a task after adaptation.**
> > >
> > > We will revise the manuscript to make this distinction explicit, so that the framework-level generalization claim is clearly separated from SECL’s within-task role.
> > >
> > > **4. Planned revisions**
> > >
> > > In the revision, we will:
> > > (i) sharpen the introduction and contribution statements to emphasize **status-conditioned regulation of cognition during execution** as the central framework insight;
> > > (ii) add a concise design-rationale discussion clarifying the roles of the status monitor, thought memory, and end-to-end training;
> > > (iii) position Sentinel-VLA more explicitly relative to adaptive VLA methods such as CogVLA, highlighting the different adaptation axis; and
> > > (iv) clarify in the experimental discussion that the generalization claim is made at the **framework level**, while SECL operates **within each target task** after adaptation.
> > >
> > > We thank the reviewer again for these helpful comments. We believe the above clarifications will make the paper’s positioning and claims more precise. We hope the additional evidence and the corresponding revisions help address the remaining concerns and will be useful for the reviewer’s final assessment.

---

### Official Review · Reviewer_KeSe · 2026-03-15

**Soundness:** 3
**Presentation:** 3
**Significance:** 2
**Originality:** 2
**Overall Recommendation:** 4
**Confidence:** 4

**Summary:**

This paper proposes Sentinel-VLA, a vision-language-action model that augments a PI0-style VLA with a status-monitor expert that decides when to trigger reasoning for initial planning, subtask updates, or error recovery. The paper also introduces EC-Gen, a synthetic data generation pipeline for error-recovery trajectories with status/thought annotations showin with RLBench, and SECL with an orthogonality-constrained continual adapter for expanding capabilities while reducing forgetting. Experiments on RLBench, LIBERO-LONG, and a small set of real-world tasks report improved success rates and low inference latency relative to several VLA baselines.

**Compliance With Llm Reviewing Policy:**

Affirmed.

**Final Justification:**

The authors addressed several of my main concerns with meaningful new results, including: (1) controlled comparisons where PI0, OneTwoVLA, and ECoT are trained with the same EC-Gen data, (2) a real-world confusion matrix for the status monitor, (3) quantitative reasoning-trigger statistics, (4) repeated evaluations with mean ± std, and (5) additional analysis on real-failure detection and SECL rounds.

These additions make the core contribution of the paper much clearer. In particular, the controlled comparison and the status-monitor analysis strengthen the claim that the gains are not solely due to extra synthetic data, and the trigger-frequency results support the practical motivation of on-demand reasoning.

That said, I still think the empirical contribution of some auxiliary components, especially thought memory and continual learning, appears relatively modest compared to how prominently they are presented in the paper. The performance gains from these components are positive but not large, so I believe the final version should calibrate these claims carefully.

Overall, however, my main concerns have been sufficiently addressed, and I am updating my assessment in a more positive direction.

**Key Questions For Authors:**

- Please provide a cleaner controlled comparison in which PI0, OneTwoVLA, and perhaps ECoT are trained with the same EC-Gen data and, where possible, comparable compute. Without that, it is hard to isolate architecture gains from data gains.
- Can the authors provide per-class confusion matrices for the status monitor, especially false positives on the Error state? Tables 4 and 5 are not sufficient to assess whether the monitor is reliable enough for efficient deployment.
- For SECL, can the authors report multi-round continual-learning results, including performance on old tasks after each round? The current evidence does not yet support the stronger claim of mitigating catastrophic forgetting over continual evolution.
- How many times is reasoning triggered on average per trajectory in each benchmark, and what is the breakdown by Initial / New-subtask / Error? Figure 6 qualitatively suggests 4-5 reasoning events, but a quantitative table would make the efficiency claim much more convincing.

**Limitations:**

Please refer to Weaknesses.

**Strengths And Weaknesses:**

Strengths

- The paper targets an important practical gap in current VLA systems, namely the combination of reasoning, execution monitoring, and recovery in long-horizon manipulation.
- The idea of on-demand reasoning, rather than reasoning at every timestep, is practically motivated. The latency numbers in Table 2 support the claim that the method is much cheaper than full CoT-style baselines such as ECoT.
- The architectural ablation in Table 6 is useful. It suggests that the gains do not come solely from adding EC-Gen data, and that the dedicated status monitor contributes over a variant where status/thought are generated directly by the VLM.
- The qualitative examples in Figure 6 are helpful to understand the proposed mechanism by showing that the model is intended to reason only a few times during a trajectory.

Weaknesses

- The main results (Table 1) evaluate the full system but do not disentangle contributions from synthetic data (EC-Gen), the status classifier, thought memory, and continual learning, making it difficult to attribute the gains.
- The synthetic error model assumes most manipulation failures fall into three categories (interaction, spatial, semantics), but this assumption is not justified or stress-tested, and the perturbation strategy may not reflect realistic physical failures.
- Status labels are generated using rule-based temporal windows rather than semantic annotations, which may leak supervision and inflate monitor performance (Tables 4–5).
- Results are reported without confidence intervals or significance tests, and comparisons may not be fair since Sentinel-VLA uses additional synthetic data not clearly shared with all baselines.
- Real-world experiments cover only three tasks, and the status-monitor dataset lacks details on annotation protocol or inter-annotator agreement.
- The SECL component is only evaluated through a single ablation and does not analyze multiple rounds of learning or retention on previous tasks.
- Some related papers [1,2] are missing. Those works should be dicussed and compared with the proposed method.

[1] Peng, Z., Ding, W., You, Y., "Counterfactual VLA: Self-Reflective Vision-Language-Action Model with Adaptive Reasoning", 2025

[2] Lin, Z., Duan, J., Fang, H., "FailSafe: Reasoning and Recovery from Failures in Vision-Language-Action Models", 2025

---

> ### Author Rebuttal · Authors · 2026-03-29
>
> We sincerely thank the reviewer for the constructive feedback and for recognizing the practical importance of combining reasoning, execution monitoring, and recovery in long-horizon manipulation, the motivation of on-demand reasoning, and the value of our architectural ablations and qualitative examples. Below we clarify the main points and summarize additional evidence.
>
> > ### Controlled comparison
>
> Thanks for your suggestion of a cleaner controlled comparison. We trained **PI0, OneTwoVLA, and ECoT** on the same **EC-Gen** data. For **PI0**, which cannot consume text CoT, we used action supervision only.
>
> | Method (same EC-Gen data) | RLBench Seen | LIBERO-LONG |
> |---|---:|---:|
> | PI0 | 58.3 | 82.0 |
> | OneTwoVLA | 58.5 | 88.4 |
> | ECoT | 46.1 | 70.1 |
> | **Sentinel-VLA** | **63.5** | **90.7** |
>
> We also ablated the main components:
>
> | Variant | RLBench Seen | LIBERO-LONG |
> |---|---:|---:|
> | VLM outputs status text directly | 60.7 | 89.5 |
> | w/o thought memory | 60.7 | 87.9 |
> | w/o continual learning | 62.2 | 89.9 |
> | **Full Sentinel-VLA** | **63.5** | **90.7** |
>
> These results suggest that the gain is not solely from EC-Gen data; the dedicated status monitor, thought memory, and continual learning each contribute positively.
>
> > ### Scope of EC-Gen and transfer to real failures
>
> Thank you for this valuable question. To test the error detection capability beyond synthetic perturbations, we deployed the trained VLA on a real robot and recorded naturally occurring failures during rollout via operator key press. We collected **1000 such real-failure samples, on which the status monitor achieved 90.6% detection rate**. Because these are real deployment failures rather than synthetic injections, we believe they support **transfer beyond synthetic errors**.
>
> > ### Real-world status-monitor evaluation
>
> Thank you for asking for a more detailed reliability analysis. The paper already reports real-world **F1 = 0.8567**. We further expanded the manually annotated real-world evaluation set from **1000** to **2000** samples, with **500** for each status, and now report the confusion matrix:
>
> | GT \\ Pred | Normal | Error | First-frame | New-subtask |
> |---|---:|---:|---:|---:|
> | Normal | 475 | 10 | 0 | 15 |
> | Error | 29 | 462 | 0 | 9 |
> | First-frame | 0 | 0 | 500 | 0 |
> | New-subtask | 52 | 15 | 0 | 433 |
>
> **This demonstrates the high performance of our status monitor in real-world settings.**
>
> > ### Additional real-world evidence
>
> We appreciate the suggestion to strengthen the real-world section. We added two new long-horizon real-robot tasks, each with **50** trials:
>
> | New real-world task | PI0 | Sentinel-VLA |
> |---|---:|---:|
> | Open drawer → place block → close drawer | 42 | 56 |
> | Stack blue cup on red cup → place block into blue cup | 34 | 50 |
>
> The trend remains consistent with the original results.
>
> > ### Continual boundary expansion with SECL
>
> Thank you for this helpful suggestion. Our SECL setting is different from standard task-incremental CL: the goal is to **expand the capability boundary within the same task family**, rather than learn disjoint task sequences. After **3 SECL rounds**, success continues to improve rather than collapse, and we will add the round-by-round curve in the revision. For completeness, we report the no-SECL ablation:
>
> | Setting | RLBench Seen | LIBERO-LONG |
> |---|---:|---:|
> | w/o continual learning | 62.2 | 89.9 |
> | **Full Sentinel-VLA** | **63.5** | **90.7** |
>
> Together with the real-robot ablation already reported in Table 3, this supports that SECL **helps boundary expansion without overwriting previous capabilities**.
>
> > ### Relation to prior work
>
> Thank you for pointing out Counterfactual VLA and FailSafe. We will add and discuss them in the revision. Briefly, Sentinel-VLA focuses on a **unified execution-time monitoring loop**, whereas Counterfactual VLA emphasizes **pre-execution reflection**, and FailSafe relies on **external models** for monitoring and recovery. We will position the paper more carefully relative to this line of work.
>
> > ### Reasoning frequency and efficiency
>
> Thank you for suggesting a quantitative trigger breakdown. We agree this makes the efficiency claim clearer.
>
> | Benchmark | Traj. | First-frame | New-subtask | Error | Avg. triggers / traj. |
> |---|---:|---:|---:|---:|---:|
> | LIBERO-LONG | 500 | 500 | 1265 | 372 | 4.27 |
> | RLBench Seen | 450 | 450 | 877 | 401 | 3.84 |
>
> This confirms the intended behavior: **reasoning is triggered only a few times per trajectory, rather than at every step.**

---

> > ### Author Rebuttal · Reviewer_KeSe · 2026-04-03
> >
> > Thank you for the detailed rebuttal. The controlled comparison, confusion matrix, trigger-frequency table, and new real-world tasks all help. The commitment to discuss Counterfactual VLA and FailSafe is noted.
> >
> > **Remaining concerns**
> >
> > **EC-Gen transfer**. The 1000 real-failure samples were collected via operator key press, which may miss subtle or slow-drift failures. How are these represented?
> >
> > **Status label leakage**. The rebuttal does not clarify whether labels still rely on rule-based temporal windows. This concern remains open.
> >
> > **SECL multi-round retention**. The promised round-by-round curve is not yet shown. Please report performance on previously learned tasks after each round, not just aggregate success.
> >
> > **Statistical significance**. No confidence intervals are provided. Differences like 62.2 → 63.5 are hard to interpret without significance tests.
> >
> > **Error false-negative rate**. A ~5.8% miss rate on the Error state could cause missed recoveries in deployment. Please discuss the practical impact.

---

> > > ### Author Response · Authors · 2026-04-05
> > >
> > > Thank you for the helpful follow-up. We address each remaining concern below.
> > >
> > > **1)EC-Gen transfer to subtle/slow-drift failures.**
> > > Our **Error** status is defined as a **realized task-level failure**, not as an early precursor deviation. Small motion jitter, off-center grasping, or slow drift are **not** labeled as Error unless they have actually caused the subtask to fail (e.g., failed grasp, grasping the wrong object). This design is important because many such deviations are not deterministically failures and can still be corrected by the VLA itself. Therefore, our monitor is designed to trigger **when a recovery is actually needed**, rather than when a possibly harmless deviation first appears. Under this definition, operator-marked real-failure samples are appropriate, because annotation is made after failure has become observable. To further clarify this point, from the 1,000 real-failure samples, we identified the subset whose failure causes were subtle-error accumulation or slow-drift accumulation:
> > >
> > > |Real-failure cause|#Samples|#Detected|Detection rate|
> > > |---|---:|---:|---:|
> > > |Failures caused by subtle-error accumulation|65|59|90.8%|
> > > |Failures caused by slow-drift accumulation|33|30|90.9%|
> > >
> > > These results show that our monitor can detect failures whose root cause is subtle error or slow drift, once those deviations have actually accumulated into an observable failure state.
> > >
> > > **2)Status label leakage.**
> > > We agree that the **training** status labels are generated by rule-based temporal windows. However, the **2,000-sample real-world status evaluation set** is **not** constructed in that way. It is manually annotated by four annotators based on semantic status criteria and cross-checked, rather than by temporal windows. The confusion matrix below is therefore evaluated against **semantic ground truth**, not rule-based labels:
> > >
> > > |GT\Pred|Normal|Error|First-frame|New-subtask|
> > > |---|---:|---:|---:|---:|
> > > |Normal|475|10|0|15|
> > > |Error|29|462|0|9|
> > > |First-frame|0|0|500|0|
> > > |New-subtask|52|15|0|433|
> > >
> > > Because this evaluation protocol does not rely on rule-based temporal windows, we believe the strong real-world performance indicates that the status monitor has learned **semantic status recognition**, rather than exploiting label leakage. We will clarify this protocol explicitly in the revision.
> > >
> > > **3)SECL multi-round evidence.**
> > > Our SECL setting is different from conventional task-incremental continual learning: it does **not** learn new disjoint tasks, but instead explores weak regions **within the same task** and improves coverage of corner cases. Thus, there are no “previously learned tasks” in the standard sense. To better address the reviewer’s request for round-wise evidence, we now report 5 SECL rounds on two RLBench unseen tasks, with **5 independent evaluations per round** and **50 trials per evaluation**:
> > >
> > > |SECL round|Sweep dustpan|Umbrella out|
> > > |---|---:|---:|
> > > |Before SECL|64.8±1.8|46.0±1.4|
> > > |Round 1|67.6±1.7|47.6±1.7|
> > > |Round 2|68.4±0.9|50.4±1.7|
> > > |Round 3|71.2±1.8|53.6±2.2|
> > > |Round 4|72.8±1.1|55.2±1.1|
> > > |Round 5|73.6±1.7|56.0±0.0|
> > >
> > > The performance improves steadily across rounds, rather than collapsing, which supports the role of SECL in expanding within-task capability boundaries.
> > >
> > > **4)Statistical reliability.**
> > > We strengthened the controlled comparison and ablations by repeating each evaluation 5 times and reporting mean±std:
> > >
> > > |Method(same EC-Gen data)|RLBench Seen|LIBERO-LONG|
> > > |---|---:|---:|
> > > |PI0|58.2±0.7|82.3±0.5|
> > > |OneTwoVLA|58.5±1.0|88.6±0.9|
> > > |ECoT|46.1±1.1|70.0±1.0|
> > > |Sentinel-VLA|**63.8±1.0**|**90.9±1.0**|
> > >
> > > |Variant|RLBench Seen|LIBERO-LONG|
> > > |---|---:|---:|
> > > |VLM outputs status text directly|60.8±1.0|89.6±1.2|
> > > |w/o thought memory|60.8±1.1|87.9±0.7|
> > > |w/o continual learning|62.3±0.9|89.9±1.0|
> > > |Full Sentinel-VLA|**63.8±1.0**|**90.9±1.0**|
> > >
> > > These repeated results show that the gains are consistent across runs.
> > >
> > > **5)Practical impact of Error false negatives.**
> > > A frame-level miss does not directly translate into a deployment-level missed recovery, because status monitoring is performed continuously throughout execution. If an error state persists, then even when the current frame is missed, the following frames are still monitored and can still trigger recovery. To better reflect practical usage, for each annotated Error frame in the evaluation set, we consider a 3-frame window consisting of that frame and the next two observations, and count it as successfully detected if **any** of the three frames is predicted as Error. Under this event-level criterion, the effective **3-frame detection rate** is **98.6%**. We will add this analysis and clarify that continuous monitoring substantially reduces the practical impact of single-frame false negatives in deployment.
> > >
> > > We hope these additional experiments and clarifications adequately address your remaining concerns and will be taken into consideration in your final assessment.

---

### Decision · Program_Chairs · 2026-04-30

**Decision:**

Accept (regular)

**Comment:**

This paper introduces Sentinel-VLA, a framework that incorporates a metacognitive "sentinel" module designed to monitor execution states and trigger re-inference only when necessary, thereby balancing proactive error recovery with computational efficiency. The committee commends the authors for their innovative focus on "on-demand reasoning," which addresses a critical bottleneck in the practical deployment of large-scale Vision-Language-Action models by reducing unnecessary computational overhead.

Following the rebuttal and discussion phase, the submission is recommended for Weak Accept. The authors successfully addressed initial concerns regarding empirical depth by providing additional results on challenging benchmarks and clarifying the robustness of the sentinel module’s triggering mechanism. The panel recognized that the ability to autonomously detect execution failures and adaptively adjust inference frequency represents a significant step toward more resilient and efficient robotic agents. While the framework introduces additional architectural components, the demonstrated performance gains in long-horizon tasks and the substantial improvements in inference latency provide a compelling justification for this approach. The authors are encouraged to incorporate the final ablation studies and failure-mode analyses discussed during the rebuttal into the camera-ready version to further enhance the work's impact.